# Groundwater origin, flow regime and geochemical evolution in arid endorheic watersheds: a case study from the Qaidam Basin, Northwest China

Yong Xiao[2], Jingli Shao[1*], Shaun K. Frape[3], Yali Cui[1], Xueya Dang[4], Shengbin Wang[5,6], Yonghong Ji[7]

[1]School of Water Resources and Environment, China University of Geosciences (Beijing), Beijing, 100083, China
[2]Faculty of Geosciences and Environmental Engineering, Southwest Jiaotong University, Chengdu, 610031, China
[3]Department of Earth and Environmental Sciences, University of Waterloo, Waterloo, N2L 3G1, Canada
[4]Xi'an Center of Geological Survey, China Geological Survey, Xi'an, 710054, China
[5]Key Lab of Geo-environment of Qinghai Province, Xining, 810007, China
[6]Bureau of Qinghai Environmental Geological Prospecting, Xining, 810007, China
[7]Lunan Geo-Engineering Exploration Institute of Shandong Province, Yanzhou, 272100, China

*Correspondence to*: Jingli Shao (jshao@cugb.edu.cn)

**Abstract.** Groundwater origin, flow and geochemical evolution in the Golmud River watershed of the Qaidam Basin was assessed using hydrogeochemical, isotopic and numerical approaches. The stable isotopic results show groundwater in the basin originates from precipitation and melt water in the mountainous areas of the Tibetan Plateau. Modern water was found in the alluvial fan and shallow aquifers of the loess plain. Deep confined groundwater was recharged by paleo-water during the late Pleistocene and Holocene under a cold climate. Groundwater in the low-lying depression of the central basin is composed of paleo-brines migrated from the western part of the basin due to tectonic uplift in the geological past. Groundwater chemistry is controlled by minerals (halite, gypsum, anhydrite, mirabilite) dissolution, silicate weathering, cation exchange, evaporation and minerals (halite, gypsum, anhydrite, aragonite, calcite, dolomite) precipitation, and varies from fresh to brine with the water types evolving from $HCO_3 \cdot Cl\text{-}Ca \cdot Mg \cdot Na$ to $Cl\text{-}Na$, $Cl\text{-}K\text{-}Na$ and $Cl\text{-}Mg$ type along the flow path. Groundwater flow patterns are closely related to stratigraphic control and lithological distribution. Three hierarchical groundwater flow systems, namely local, intermediate and regional, were identified using numerical modelling. The quantity of water discharge from these three systems accounts for approximately 83%, 14% and 3%, respectively, of the total groundwater quantity of the watershed. This study can enhance the understanding of groundwater origin, circulation and evolution in the Qaidam Basin as well as other arid endorheic watersheds in northwest China and elsewhere worldwide.

**Keywords.** Groundwater origin; Hydrogeochemistry; Groundwater flow pattern; Arid area; Qaidam Basin

## 1 Introduction

Closed basins in arid and semiarid areas (e.g. the Great Artesian Basin and Murray Basin in Australia, Minqin Basin and Qaidam Basin in China, Death Valley in United States) have been the focus of attention due to their water scarcity, fragile

ecology and rich mineral resources related to salt lakes (Edmunds et al., 2006; Lowenstein and Risacher, 2009; Love et al., 2013; Shand et al., 2013; Stone and Edmunds, 2014; He et al., 2015; Cartwright et al., 2017; Love et al., 2017; Priestley et al., 2017a; Xiao et al., 2017). Groundwater plays a vital role in water supply, ecology maintenance, transportation of chemical components, as well as the formation of oil, gas reservoirs and mineral resources in these basins (Toth, 1980; Jiang et al., 2014; Jiao et al., 2015; Xiao et al., 2017). Understanding the regimes of groundwater recharge, flow and hydrogeochemical evolution is essential to maintain proper management, and implement sustainable utilization of groundwater and mineral resources, as well as maintaining the ecological environment (Cartwright et al., 2010a; Herrera et al., 2016).

In the arid northwest of China there are many closed basins such as the Tarim, Qaidam, Junggar, and Minqin Basin, in which the low-lying discharge areas are occupied by saline lakes, salt playas and salt crusts. The Qaidam Basin (Figure 1a, b), the largest closed basin of the Tibetan Plateau, has the most plentiful number of salt lakes and salt playas and almost all varieties of salt deposits (Zheng et al., 1993), as well as rich oil and gas reservoirs (Tan et al., 2011; Ye et al., 2014). Considerable research has been conducted to provide support for water supply and mineral resource exploitation in the basin (Chen and Bowler, 1986; Vengosh et al., 1995; Lowenstein and Risacher, 2009; Li et al., 2010; Tan et al., 2011; Hou et al., 2014; Ye et al., 2014; Chen et al., 2017). However, most of the previous studies focused on the groundwater in the piedmont areas (Wang and Ren, 1996; Wang et al., 2010; Zhang, 2013; Hou et al., 2014; Su et al., 2015; Xu et al., 2017), material source of salt lakes (Vengosh et al., 1995; Lowenstein and Risacher, 2009; Tan et al., 2011; Chen et al., 2015), and the evolution of salt lakes (Chen and Bowler, 1986; Chen et al., 2017). The systematic understanding of regional groundwater regimes is still inadequate. This would limit the comprehensive planning and management of groundwater and salt lake mineral resource exploitation, and finally make it difficult to safeguard the circulation of the groundwater system and maintain the eco-environment at balance. Therefore, several attempts have been made to understand the regional groundwater regimes (Tan et al., 2009; Gu et al., 2017; Xiao et al., 2017), but very little research reported the circulation and evolution of groundwater from the mountain pass area to the central terminal lake area due to the notable difficulties to move through and access the swamps on the lacustrine plain. This would greatly limit the full understanding of the role of hydrogeological processes in the basin.

Hydrogeological survey efforts have been undertaken in the Golmud River watershed of the basin since 2015, and have developed a better understanding of regional hydrogeological conditions. The main objective of this study is to assess the regional hydrogeological regime of closed basins in the arid northwest of China, using the Golmud River watershed as a case study. To achieve this aim, a comprehensive approach using environmental isotopes ($^2$H, $^{18}$O, $^3$H, $^{13}$C, $^{14}$C) and hydrochemistry coupled with numerical simulation was performed. Stable hydrogen and oxygen can provide valuable information on the origin and recharge environment of groundwater, and radioactive isotopes such as $^3$H, $^{14}$C record the residence time of groundwater (Cartwright et al., 2007; Awaleh et al., 2017; Huang et al., 2017). Hydrochemical composition has recorded the recharge water characteristics, hydrostratigraphic information, geochemical interaction, and other processes along the groundwater flow path (Redwan and Moneim, 2015; Verma et al., 2016; Love et al., 2017), and

thus can be used to track groundwater evolution. Numerical simulation of groundwater flow is an essential tool to synthesize hydrogeological information and reveal groundwater flow patterns (Bredehoeft and Konikow, 2012; Anderson et al., 2015; Tóth et al., 2016). The combination of these approaches is robust to reveal groundwater origin, flow regimes, renewability, hydrochemical evolution, inter-aquifers mixing, as well as surface water and groundwater interactions, etc., in basins with complex hydrogeology or sparse monitoring data, and has been successfully applied in many basins such as the Great Artesian Basin and Murray Basin in Australia, Michigan Basin in US, Minqin Basin and Ordos Plateau in China, Stampriet Basin in Africa (Edmunds et al., 2006; Banks et al., 2010; Love et al., 2013; Stone and Edmunds, 2014; Su et al., 2016; Cartwright and Morgenstern, 2017; Love et al., 2017; Petts et al., 2017; Priestley et al., 2017b).

The specific aims of the present study are to: (1) identify the recharge source of groundwater, (2) assess the regional groundwater chemistry characteristics, (3) determine the controlling mechanisms of hydrogeochemistry, (4) delineate regional groundwater flow patterns, (5) and ultimately establish systematic regional groundwater regimes from the mountain pass to the terminal lake in the typical Golmud watershed of Qaidam Basin. This study would provide insights into the origin, recharge environment, flow regime and geochemical evolution of regional groundwater in arid endorheic watersheds of Qaidam Basin, and provide reference for other arid closed basins in northwest China as well as similar endorheic watersheds worldwide.

## 2 Study area

The Qaidam Basin is a large closed basin located on the north-eastern margin of the Tibetan plateau, surrounded by the Qilian Mountains to the north, the Kunlun Mountains to the south, and the Altun Mountains to the west (Figure 1b). The study area, Golmud River Watershed (GRW), is located in the southern part of the Qaidam Basin hosting the second largest river, the Golmud River, running from the Kunlun Mountains in the south to the low-lying depression in the north central part of the Basin (Figure 1c). The Qarhan salt lake is the largest salt lake in China located at the northern margin of the GRW, adjacent to the Golmud River's terminal Lake Dabusun. The third largest city on the Tibetan plateau, Golmud City, is also located in the GRW.

The outcropping stratigraphy of the GRW ranges from Proterozoic to Quaternary in age. The Quaternary strata are found in the mountainous areas to the south. These strata have undergone magmatic activity, uplift, tectonic movements, as well as intense weathering, resulting in massive material sources of sediments to the basin. The Quaternary deposits have thicknesses ranging from hundreds of meters in the piedmont area to thousands of meters in the low-lying depression (basin center) (Zheng et al., 1993; Chen et al., 2017). Field surveys found that salt crusts are formed on the ground surface in locations near the terminal areas of streams. Core drilling records also show many salt-bearing deposits such as halite, calcium, sulfate, sodium sulfate were observed throughout the strata (Chen and Bowler, 1986). The regional Quaternary aquifers in the basin vary from single unconfined gravel and sand layers with hydraulic conductivity (K) greater than 50 m/d in the alluvial fan to multi-layers of silt and clay with hydraulic conductivity (K) ranging from 0.1 m/d to 0.001 m/d in the

low-lying depression (basin center). Three continuous aquitards (clay layers) are found in the basin at depths of 60 m, 290 m and 450 m, respectively (Figure 8), which have significant influences on confining groundwater flow (Shao et al., 2017).

The climate in the GRW is extremely variable, both spatially and temporally. Precipitation in the Kunlun Mountains is more than 200 mm per year, but less than 50 mm in the basin, and also presents a gradual decreasing trend from the piedmont area to the low-lying central depression. The potential evaporation is extremely high ($>$2600 mm per year). This hyper-arid climate results in aquifers in the basin that do not obtain effective recharge from the local precipitation. Groundwater in the basin is mainly recharged by Golmud River seepage through the riverbed in the alluvial fan and bedrock lateral inflow at the southern mountain front, and flows from the alluvial fan in the south to the basin center in the north (Figure 1c). Much of groundwater overflows as springs at the front of the alluvial fan due to the fining of sediments in the aquifers downdip. The depth to groundwater is less than 3 m in most areas from the front of alluvial fan to the basin center, resulting in significant potentially evaporate loss of groundwater. The regional groundwater finally discharges to the terminal lake, and undergoes large evaporate loss (Shao et al., 2017).

Based on the terrain, sediments and hydrogeological condition, the study area can be divided into 5 zones. Zone 1 is the Kunlun mountainous area, and Zone 2 is the alluvial fan plain at the Kunlun piedmont. Zones 3, 4 and 5 occur on the loess plain, where Zone 3 is the main groundwater overflow (discharge) zone of the watershed, and Zone 5 is the terminal lake zone (low-lying depression of the watershed) with salt crusts and playa, Zone 4 is the transition zone (middle-lower stream area of the watershed) between Zone 3 and 5.

## 3 Materials and Methods

### 3.1 Hydrochemical and isotopic sampling and analytical methods

A total of 228 water samples were collected from GRW in 2015 and 2016, including 180 groundwater samples and 48 surface water samples (42 river water and 6 lake water samples) (Figure 1c). Groundwater samples were collected from both shallow phreatic aquifers and deep confined aquifers. Surface water samples were obtained along the Golmud River, as well as from Lake Qarhan, Lake Dabusun and other small lake in the low-lying depression area (basin center). In addition, one snow (snowmelt water) sample, 8 precipitation samples and 90 brine water samples (groundwater) with hydrogen and oxygen stable isotope data were obtained from China's stable isotope geochemistry database (http://210.73.59.163/isogeochem/). The location of the snow and precipitation samples are shown on Figure 1c. The detailed locations of the 90 brine water samples are not known, but it is known that all of these samples were collected from the Qarhan salt playa and Bieletan salt playa (Figure 1c).

For groundwater sampling, all wells and boreholes, except those that were artesian, were pumped for several well volumes to remove the stagnant water in the wells and boreholes, and monitored until the electrical conductivity (EC) of the pumping water was stable. The sampling procedure followed is described in Huang et al. (2016) and Chen et al. (2011). Samples for major element analysis were collected in two 250 ml high density polyethylene bottles after filtration using 0.45 μm filter

membranes (Huang et al., 2016). Samples for tritium ($^3$H) and stable isotopes ($^2$H, $^{18}$O) analyse were collected in 500 ml and 50 ml glass bottles, respectively, that were filled to overflowing after rinsing and were sealed tightly. $^{13}$C and $^{14}$C samples were collected by adding $BaCl_2$ and $CO_2$-free NaOH to 120 L groundwater at pH=12, obtaining $BaCO_3$ for dissolved inorganic carbon (DIC) analysis (Chen et al., 2011). The method used eliminates contact with the atmosphere in order to avoid $CO_2$ atmospheric contamination.

Parameters such as the water temperature (T), pH, EC were measured in the field with an in situ multi-parameter instrument (Multi 350i/SET, Munich, Germany), and redox potential (Eh) was also determined in situ using a portable ORP tester (CLEAN ORP30 Tester, California, US). Major chemistry and isotopes ($^2$H, $^{18}$O, $^3$H, $^{13}$C and $^{14}$C) of the sampled water were analyzed at the Laboratory of Groundwater Sciences and Engineering in the Institute of Hydrogeology and Environmental Geology, Chinese Academy of Geological Sciences (Shijiazhuang, Hebei Province, China). Major cations ($K^+$, $Na^+$, $Ca^{2+}$, $Mg^{2+}$) were measured by inductively coupled plasma-mass spectrometry (Agilent 7500ce ICP-MS, Tokyo, Japan). Total dissolved solids (TDS) and $HCO_3^-$ were determined by gravimetric analysis and acid-base titration, respectively. $Cl^-$, $SO_4^{2-}$ were analyzed using spectrophotometry (PerkinElmer Lambda 35, Waltham, MA, USA). The ionic charge balance of all samples were within 5% difference. $\delta^{18}O$, $\delta^2H$, $\delta^{13}C$ were measured by isotope ratio mass spectrometry using a Finnigan MAT 253, and $\delta^{18}O$, $\delta^2H$ were reported relative to the Vienna Standard Mean Ocean Water (VSMOW) standard, and $\delta^{13}C$ was reported relative to the Vienna Pee Dee Belemnite (VPDB). The analytical errors are ±0.2‰ for $\delta^{18}O$, ±1.0‰ for $\delta^2H$ and ±0.5‰ for $\delta^{13}C$. The tritium content was determined using electrolytic enrichment and liquid scintillation technique (Chen et al., 2011) with the precision of ±0.3 TU. The activity of $^{14}$C was analyzed by liquid scintillation counting (1220 Quantulus), and expressed as a percentage of modern carbon (pMC) with the precision of ±0.3% (Su et al., 2018).

## 3.2 Two-dimensional groundwater flow numerical simulation

It is assumed that the variation of density and viscosity of waters could be neglected for calculations involving most of the flow system (Zone 1~4). For simplicity, groundwater in the terminal lake zone (Zone 5) is also regarded as being mainly driven by gravity. Thus the equation governing variably saturated groundwater flow is as follows (Richards, 1931):

$$\frac{\partial}{\partial t}\emptyset S = \text{div}[K\nabla h]$$

Where $\phi$ is the porosity, S is the liquid saturation, K is the hydraulic conductivity (m/d), h is the hydraulic head (m). The TOUGH2 code (Transport Of Unsaturated Groundwater and Heat), which has quite robust simulation capabilities, is used to numerically solve this equation (Pruess et al., 1999).

The cross section parallel to the main direction of groundwater flow in GRW (Figure 1 & 8) was selected for the 2-D flow simulation. This section starts at the mountain pass and ends at the Terminal Lake Dabusun, with an approximate length of 100 km. Boundaries were specified according to the hydrogeology condition. The southern lateral boundary and top boundary in the alluvial fan are defined as given flux boundaries, and the bottom boundary of the section is regard as a zero flux boundary. The springs and evaporation are set as mixed boundaries. The lake boundary in the basin center is specified as a given head boundary.

An irregular discretization was conducted vertically to capture the variation of water table near the ground surface and also implement an efficient simulation. Cells are presented with the minimum thickness of 0.1 m near the ground surface and gradual increasing thickness downward, with a maximum thickness of about 80 m. Equal discretization was applied in the horizontal direction with a horizontal size of 1000 m for one cell. The initial permeability of various lithologies are specified based on the borehole drilling records and pumping test results, with the $K_h$ (horizontal hydraulic conductivity) in the range of $10^2 \sim 10^{-3}$ m/d and anisotropy ratio $K_h/K_v$=5~10 ($K_v$ is vertical hydraulic conductivity) (Shao et al., 2017). In this study, the model is used to present the flow pattern under equilibrium conditions, thus the recharge rates and hydraulic heads are given according to the annual average values. Evaporation was modelled using a newly developed method described by Hao et al. (2016), and the initial potential evaporation specified is 2600 mm per year. Springs are simulated using the DELV module in TOUGH2, and the productivity index (PI) specified in DELV module is calculated using the following equation (Pruess et al., 1999):

$$PI = \frac{2\pi(k\Delta z)}{\ln\left(\sqrt{A/\pi}/r\right) + s - 1/2}$$

Where $\Delta z$ is the layer thickness (m), A is the grid block area (m$^2$), r is the spring radius (m), s is the skin factor. Annual average observed hydraulic heads are used as natural constraints for the model calibration.

## 4 Results

### 4.1 Hydrochemistry of surface waters and groundwaters

The statistical summaries of chemical analysis results for surface water and groundwater are presented in Table1. River waters (RW) from the mountain pass (Zone 2) to the low-lying depression (Zone 5) are slightly alkaline with a range in pH

values from 7.94 to 9.45. Fresh lake water (FLW) L1, which was sampled from the fresh lake (relative fresh compared to other salt lakes) recharged directly by river water in the low-lying depression (Zone 5), is also slightly alkaline with a pH value of 8.98. Samples from the salt lakes such as Lake Qarhan, Lake Dabusun are slightly acidic with values range from 6.03 to 6.28. Groundwater in the study area is neutral to slightly alkaline. The shallow phreatic groundwater (SGW) shows

an evolving trend from slightly alkaline to slightly acidic along the flow path with the pH varying from 9.34 to 6.03. However, deep confined groundwater (DGW) samples are all slightly alkaline with pH values ranging between 7.83 and 8.69. The redox potential (Eh) of SGW are in the range of 123-162 mV from alluvial fan to middle lower stream area (Zone 2, Zone 3 and Zone 4), suggesting an oxidation condition. The Eh values of DGW vary from 153 mV to 40 mV along the flow path (Zone 3 to Zone 4), indicating the redox condition gradually evolves from a state of oxidation to reduction (Figure 3e).

Surface water and groundwater present distinct major solute chemistry across the study area. As shown in Table 1, the concentration of ions in RW demonstrates an increase along river flow paths, with a TDS values varying from 393 mg/L to 2,319 mg/L. The TDS value of FLW (L1) is much higher than that of RW in the low-lying depression (Zone 5), with the TDS value of 10,937 mg/L. While the salt lake waters (SLW) have extremely high TDS values ranging from  339,098 mg/L to 403,758 mg/L. The dominant ions of RW are $HCO_3^-$ and $Na^+$ with the concentration range of 184-215 mg/L for $HCO_3^-$

and 63-92 mg/L for $Na^+$, respectively, in the alluvial fan area (Zone 2), and gradually evolve to $Cl^-$ and $Na^+$ with the concentration range of 655-1,776 mg/L for $Cl^-$ and 438-996 mg/L for $Na^+$, respectively, in the low-lying depression (Zone 5). FLW (L1) has the same dominant ions with RW in the low-lying depression (Zone 5), but with higher concentration of 5,912 mg/L for $Cl^-$ and 2,957 mg/L for $Na^+$. SLW is dominated by $Cl^-$ and $Mg^{2+}$ with the concentration range of 276,849 mg/L to 285,780 mg/L for $Cl^-$ and 99,500 mg/L to 100,240 mg/L for $Mg^{2+}$, respectively. Overall, the surface water types evolve from

$HCO_3 \cdot Cl \cdot Ca \cdot Mg \cdot Na$ type in the alluvial fan area (Zone 2) to Cl-Na, Cl-K-Na and Cl-Mg type in the low-lying central depression (Zone 5) (Figure 2a).

Groundwater shows a similar hydrochemical evolution along the flow path. The average TDS values vary from 618 mg/L to 32,029 mg/L for SGW and from 547 mg/L to 1,401 mg/L for DGW from the upstream area (Zone 2) to the middle-lower stream area (Zone 4). DGW is much fresher when contrasted with the SGW at the same location (Figure 2c). There is

essentially no difference in TDS between SGW and DGW from the central depression (Zone 5) with the values ranging from 336,229 mg/L to 361,200 mg/L for SGW and 370,940 mg/L for representative DGW (Table 1). Groundwater in the alluvial fan area (Zone 2) is dominated by $HCO_3^-$, $Cl^-$ and $Na^+$ with the concentration ranging from 89 mg/L to 309 mg/L for $HCO_3^-$, from 90 mg/L to 437 mg/L for $Cl^-$, and from 79 mg/L to 232 mg/L for $Na^+$, respectively. To the middle-lower stream area (Zone 4), the dominant ions vary to $Cl^-$ and $Na^+$ for both SGW and DGW. The mean concentration of $Cl^-$ is 11,550 mg/L for

SGW and 263 mg/L for DGW, and the average concentration of $Na^+$ is 10,464 mg/L for SGW and 407 mg/L for DGW. All groundwaters including SGW and DGW in the basin center (Zone 5) are dominated by $Cl^-$, $Na^+$ and $Mg^{2+}$. SGW has the concentration ranging from 215,561 mg/L to 227,451 mg/L for $Cl^-$, from 12,388 mg/L to 35,713 mg/L for $Na^+$, and from 53,480 mg/L to 64,860 mg/L for $Mg^{2+}$. The concentration of representative DGW is 222,404 mg/L for $Cl^-$, 32,378 mg/L for $Na^+$, and 57,079 mg/L for $Mg^{2+}$. Overall, the water types of both SGW and DGW evolve from $HCO_3 \cdot Cl \cdot Ca \cdot Mg \cdot Na$ type in

the upstream area (Zone 2) to Cl-Na type in the middle-lower stream area (Zone 4), and eventually to Cl-Mg type in the low-lying depression (Zone 5) (Figure 2b).

## 4.2 Stable and radio isotopes

The statistical summary of isotope results for precipitation, river water, lake water and groundwater can be found on Figure 3 and Table 2. The representative snowmelt water in the Kunlun Mountainous area (Zone 1) has a δD value of -77.0 ‰ and δ$^{18}$O value of -11.9 ‰. The δD and δ$^{18}$O values of precipitation in the mountainous area (Zone 1) are in the range of -85.3 to -71.6 ‰ and -10.9 ‰ to -9.3 ‰, with an average value of -75.2 ‰ and -10.0 ‰, respectively. The δD values of precipitation in the alluvial fan (Zone 2) range from -68.1 ‰ to -66.2 ‰ with the average value of -67.2 ‰ and δ$^{18}$O values ranging from -10.1 ‰ to -9.7 ‰ with an average value of -9.9 ‰. The enrichment of stable hydrogen and oxygen isotopes in precipitation from the mountainous area to the basin reflects secondary evaporation effect of precipitation in arid inland areas (Clark and Fritz, 1997). The δD and δ$^{18}$O values of river water vary from -75.7 ‰ to -46.7 ‰ and between -11.1 ‰ and -4.8 ‰, respectively, showing a gradual enrichment trend along the river flow path. Fresh and salt lake waters are all significantly enriched in heavy isotopes with values ranging from -27.0 ‰ to -4.0 ‰ for δD and from 0.4 ‰ to 4.5 ‰ for δ$^{18}$O. As shown on Figure 3 a and b, the SGW from the alluvial fan (Zone 2) to the middle-lower stream area (Zone 4) show an overall gradual enrichment trend along the flow path. In contrast, the DGW shows a significant depletion trend from the south to the north. Groundwater at different depths in the low-lying depression (Zone 5) are all brines with δD values ranging from -66.0 ‰ to -2.0 ‰ and the δ$^{18}$O values ranging between -10.8 ‰ and -0.6 ‰, demonstrating relative enriched characteristics in contrast with the fresher groundwater in the upstream areas.

The $^{3}$H values range from 56.3 TU to 12.1 TU in the SGW and from 25.7 TU to <1 TU in the DGW along the groundwater flow path (Figure 3d). The $^{14}$C activities in SGW vary from 57.9 pMC to 11.9 pMC, and in DGW range from 49.15 pMC to 0.7 pMC along the flow path (Figure 3c). The spatial distributions of $^{3}$H and $^{14}$C results indicate increasing residence times for groundwaters in the aquifers from the south to north. While one shallow phreatic groundwater sample (G178) in the low-lying depression (Zone 5) was observed with relative high $^{3}$H content (18.9 TU), this may be caused by the rapid infiltration of surface water in flood period.

Groundwater in the alluvial fan (Zone 2) has a high tritium content ranging from 20.0 TU to 56.3 TU with the average value of 35.5 TU, indicating recharged by modern water with the age less than 60 years. Shallow groundwater in the overflow zone (Zone 3) and middle lower stream area (Zone 4) also has a relative high tritium content in the range of 12.1-25.7 TU with an average value of 21.1 TU for Zone 3 and 14.4-17.5 TU with an average value of 16.0 TU for Zone 4, and the representative shallow phreatic water adjacent to the salt lake (Zone 5) also shows a high tritium content of 18.9 TU, presenting modern water isotopic signatures. This may be caused by the mixture with the infiltrating modern surface water. DGW in the overflow zone (Zone 3) and middle lower stream area (Zone 4) are with tritium content ranging from <1 TU to 10.1 TU for Zone 3 and <1 TU to 4.1 TU for Zone 4. The elevated tritium determined from several deep confined water is most likely caused by mixtures with shallow phreatic water in the borehole, therefore, they cannot be used for groundwater

age determination. Most DGW samples have a tritium content less than 1 TU, indicating they are not influenced by mixing with shallow groundwater in the boreholes. The age of these tritium free DGW can be estimated using the radiocarbon activity.

Radiocarbon activity of groundwater can be significantly influenced by geochemical reactions (e.g. carbon minerals dissolution, isotopic exchange processes) during subsurface infiltration and in the aquifers (Cartwright et al., 2010b). It is therefore essential to correct the $^{14}C$ activity on the total dissolved inorganic carbon (TDIC) before using it for groundwater age estimation. Many model such as statistical models, geochemical models, and mixing models were proposed for $^{14}C$ activity correction. Most of the models are of limited interest due to the assumptions of fully closed system or open system, simplification or even fully ignorance of geochemical reactions beyond the recharge area. Carbon-13 based model is a good approach to correct the influence of geochemical reactions on $^{14}C$ activity on TDIC, and suitable for both open and closed system. The measured apparent $^{14}C$ activity ($^{14}C_{uncorr}$) on TDIC were corrected using $\delta^{13}C$ as following (Clark and Fritz, 1997):

$$^{14}C_{corr} = {^{14}C_{uncorr}} \frac{\delta^{13}C_{rech} - \delta^{13}C_{carb}}{\delta^{13}C_{TDIC} - \delta^{13}C_{carb}}$$

Where $^{14}C_{corr}$ is the corrected $^{14}C$ activity on TDIC, $\delta^{13}C_{TDIC}$ is measured $\delta^{13}C$ ratio on TDIC, $\delta^{13}C_{rech}$ is the assumed initial $\delta^{13}C$ ratio, and $\delta^{13}C_{carb}$ is the $\delta^{13}C$ ratio of carbonate being dissolved.

Groundwater in the study area is mainly recharged by Golmud River seepage in the upper alluvial fan located near parts of the Gobi desert where there is a lack of vegetation. The $^{14}C$ activity and $\delta^{13}C$ ratio on TDIC of the water would not be changed when infiltrating though the unsaturated zone. Thus, the $\delta^{13}C_{rech}$ ratio should be equal or close to the atmospheric value (-6.4 ‰). $\delta^{13}C_{carb}$ is close to 0 ‰ (Clark and Fritz, 1997). Only some of the tritium free DGW samples in Zone 3 and Zone 4 have measured $\delta^{13}C$ data, and these were selected to calculate groundwater age using the aforementioned $\delta^{13}C$ correction approach. The age of DGW in Zone 3 and Zone 4 ranges from 2,264 years to 20,754 years along the flow paths. Due to the absence of radiocarbon data, the age of paleo groundwater in Zone 5 cannot be calculated, but it is certain that the age is more than 20,000 years which was deduced from the oldest age of groundwater in Zone 4 (20,754 years).

### 4.3 2D groundwater flow modelling

The groundwater flow model was calibrated using annual average hydraulic heads from 63 different shallow wells measured in 2015 along the cross section (not shown on Figure). The calibration shows a good match between simulated and observed hydraulic heads as demonstrated in Figure 4. The comparison results show that the fit to observed hydraulic heads is better in the loess plain (including Zone 3, 4, 5) with the maximum deviation less than 0.8 m, while relative poor in the alluvial fan (Zone 2) with the maximum deviation less than 5 m. The deviation in the loess plain is mainly caused by the heterogeneity and anisotropy in lithology (Gu et al., 2017). The relative large deviation in results within the alluvial fan is most likely attributed to the steep hydraulic gradient (Islam et al., 2017) and larger seasonal fluctuation of hydraulic heads. Over the

whole study area, the RMSE (Root Mean Squared Error) is only 1.57 m, therefore, the calibrated model can be used to reveal the groundwater flow pattern.

The estimated hydraulic parameters are shown in Table 3. The estimated values of $K_h$ are 56.3 m/d for gravel sand, 13.7 m/d for sand, 0.62 m/d for sandy silt, 0.13 m/d for silt and 0.001 m/d for clay. The anisotropy ratio of $K_h/K_v$ was estimated as 10 for gravel sand and sand, and 5 for sandy silt, silt and clay. These parameters are effective values under the assumption of homogeneity in each layer. The water budget analysis indicates a dynamic equilibrium state with the equilibrium difference of 0.62%. Springs are the dominant discharge form, followed by evaporation and lake discharge, accounting for 76.81%, 22.44% and 1.37%, respectively.

## 5 Discussion

### 5.1 Water provenance and recharge characteristics

The δD and δ¹⁸O isotope analysis results for different water types are shown on Figure 5a in relation to the Global Meteoric Water Line (GMWL: δD=8×δ¹⁸O+10) (Craig, 1961). The Golmud Watershed Local Meteoric Water Line (LMWL: δD=6.98×δ¹⁸O+9.6) (Wang, 2014) and Golmud Watershed Local Evaporation Line (LEL: δD=4.09×δ¹⁸O+28.1, R²=0.94), which is the linear regression line of river and lake water in the study area, are also shown on Figure 5a. The slope and intercept of the LMWL (6.98 and 9.6) are lower than those of the GMWL (8 and 10) as a result of secondary evaporation which occurred during precipitation, reflecting the arid climatic characteristics of the study area (Dogramaci et al., 2012; Wang et al., 2017).

As shown in Figure 5a, most of the surface water and groundwaters in the study area are situated close to the GMWL and LMWL, indicating a meteoric origin. However, the spatial distribution of precipitation is extremely uneven. Most of precipitation occurred in the Kunlun mountainous area to the south. Precipitation in the basin is very limited (annual rainfall less than 50 mm) and in this area there is little effective recharge to the aquifers (Xiao et al., 2017). Thus, surface water and groundwater in the study area mainly originates from meteoric water (including precipitation and snowmelt) in the mountainous areas. River water (δD: -75.4~-64.8 ‰, δ¹⁸O: -11.1~-9.3 ‰) and groundwater (δD: -65.0 ‰, δ¹⁸O: -9.7 ‰) in the mountainous area (Zone 1) have nearly similar stable water isotopic signatures with precipitation (δD: -85.3~-71.6 ‰, δ¹⁸O: -10.9~-9.3 ‰) and snowmelt water (δD: -77.0 ‰, δ¹⁸O: -11.9 ‰) values from the Kunlun mountainous area (Zone 1), indicating their direct recharge relationship (Figure 5a). River waters flow towards the northern low-lying depression of the central basin, and show a gradual enrichment trend due to intensive evaporation. Lake waters sampled from the low-lying depression (Zone 5) have the most enriched stable water isotope values, and lie at the end of LEL defined by progressive evaporative enrichment of river water samples (Figure 5b).

The δD and δ¹⁸O values of the SGW and DGW demonstrate different varying trends along the groundwater flow path. The SGW shows a gradual positive enrichment trend in heavy isotopes along the LEL (Figure 5c), implying the influence of

evaporation. For the alluvial fan (Zone 2), the $\delta D$ and $\delta^{18}O$ values of groundwater are very similar to that of river water in the alluvial fan (Zone 2) and groundwater in the mountainous area (Zone 1) (Table 2), indicating groundwater in the alluvial fan (Zone 2) is recharged directly by the seepage of river water and lateral inflow from the mountainous area, and out of the influence of evaporation. This is confirmed by similarities in major chemical composition (Figure 2). The similar stable isotopic values also signify groundwater in the alluvial fan (Zone 2) has a short residence time, which is corroborated by elevated $^3H$ (20.0~56.3 TU, mean value of 35.5 TU) indicating the residence time is less than 60 years based on $^3H$ data (Xiao et al., 2017). SGW in the overflow zone (Zone 3) and the middle-lower stream area (Zone 4) has relative higher stable water isotope values compared with that in the alluvial fan and plots along the LEL, indicating SGW is influenced by evaporation from the overflow area (Zone 3) to the downstream. SGW in these two zones (Zone 3 & 4) also presents similar stable hydrogen and oxygen isotopic signatures as the river waters in the same area (Table 2), implying SGW has a very close hydraulic relationship with the rivers. The $^3H$ values of the SGW in Zone 3 and Zone 4, range from 12.1 TU to 25.7 TU and from 14.4 TU to 17.5 TU, respectively, with the mean value of 21.1 TU and 16.0 TU, suggesting that SGW in these two zones contain a large component of modern water or mixtures of old and modern water.

DGW in the overflow zone (Zone 3) and the middle-lower stream area (Zone4) are observed to have a completely opposite evolution trend in that the $\delta D$ and $\delta^{18}O$ values become more depleted along the groundwater flow path (Figure 5d). The depleted nature of the $\delta D$ and $\delta^{18}O$ values may have two interpretations: (1) these aquifers have another recharge region where rainfall with low $\delta D$ and $\delta^{18}O$ values occurs; or (2) the groundwater is ancient water recharged under colder climatic conditions (Chen et al., 2012; Awaleh et al., 2017). If (1) is the reason that groundwater would be more depleted in $\delta D$ and $\delta^{18}O$ along the groundwater flow paths, it is difficult to construct a mixing model that would supply more depleted waters along a deep flow path. According to the groundwater age estimated using $^{14}C$ activity, the DGW in Zone 3 and Zone 4 was recharged from 2,264 years B.P. to more than 20,754 years B.P. (Holocene to late Pleistocene), which was a period when the climate changed from cold and wet condition (30,000 years B.P. to 17,000 years B.P.) to warm and dry condition (14,000 years B.P. to present) (Zhang et al., 2011). Consequently, it is believed that the depleted $\delta D$ and $\delta^{18}O$ waters in the DGW were recharged by paleo water under a colder climate relative to present day. Similar findings were reported in the adjoining Nomhon watershed of the Qaidam Basin (Xiao et al., 2017). Additionally, this is consistent with the paleo-climate findings recorded using groundwater data from other basins of NW China (He et al., 2015; Huang et al., 2017).

Groundwater in the low-lying depression area (Zone 5), regardless of depth, are all brines with TDS values greater than 100,000 mg/L. Given the tectonic activity and depocenter migration within the Qaidam Basin over geological history (Chen and Bowler, 1986; Zhang, 1987), groundwater in the low-lying depression area (Zone 5) has a large component of paleo-brines migrated from western Qaidam Basin due to the uplift in the past (Huang and Han, 2007). According to the $^{14}C$ age of DGW in Zone 4, the deduced age of DGW in the low-lying depression (Zone 5) is more than 20,000 years. SGW was observed with high $^3H$ content (18.9 TU) adjacent to the Terminal lake (G178) (Figure 1 and Table 2), indicating mixing with leakage of modern surface water. As shown in Figure 5a, most groundwaters in the basin center show a considerable

deuterium excess, indicating that the original precipitation waters experienced considerable evaporation and vapor re-equilibration during recharge (Clark and Fritz, 1997).

## 5.2 Mechanisms controlling hydrochemistry

Generally, the composition of natural groundwater is primarily controlled by the chemical composition of recharge waters, water-aquifer matrix interaction, and groundwater residence time (Redwan and Moneim, 2015; Verma et al., 2016). As exhibited in the diagrams between TDS vs. $Na^+/(Na^++Ca^{2+})$ and $Cl^-/(Cl^-+HCO_3^-)$ (Figure 6), the major mechanisms controlling groundwater chemistry are water-rock interaction and evaporation-mineral precipitation processes (Gibbs, 1970). Water-rock interaction processes dominant the controls on groundwater chemistry at all depths in alluvial fan (Zone 2) due to the great depth and the negligible impact of evaporation. For the overflow zone (Zone 3) and the middle-lower stream area (Zone 4), the governing mechanisms for SGW change from water-rock interaction to evaporation-mineral precipitation due to the gradual decrease of groundwater depth and recharge inputs from waters having undergone the influence of intensive evaporation in that part of the basin. Nearly all DGW in this part of the flow system are controlled by water-rock interaction. Two DGW samples are observed to plot in the evaporation-crystallization domain (Figure 6). This is due to a high TDS and over-saturation of evaporative minerals (such as aragonite, calcite and dolomite) in the groundwater resulting in mineral precipitation (crystallization). For the low-lying depression (Zone 5), evaporation has a significant influence on the chemistry of SGW, and crystallization (precipitation) of many mineral phases is the primary geochemical process controlling both the SGW and DGW chemistry.

In order to further constrain the sources of solute in groundwater, the relationships between various ions are compared (Figure 7). The relation of $Na^+$ vs. $Cl^-$ shows both SGW and DGW from the piedmont to the middle-lower stream area (Zone 2, 3, 4) are plotted along the 1:1 line (Figure 7a), suggesting that halite dissolution is potentially a primary process/source of $Na^+$ and $Cl^-$ mineralization in groundwater. The calculated results of halite saturation index ($SI_{halite}<0$) (Table 4) confirm that halite minerals of the aquifer matrix could be readily available to the groundwater. In addition, core drilling demonstrated that evaporate salts such as halite, calcium sulfate and sodium sulfate are widespread in the aquifer materials, and can provide the solute source. Some of the SGW in Zone 4 are observed to have excess $Na^+$ relative to $Cl^-$ (Na/Cl ratios equal to 1.2-3.8), while groundwater in Zone 5, regardless of the depth, shows deficiency of $Na^+$ with respect of $Cl^-$ (Na/Cl ratios equal to 0.08-0.26), implying the existence of some other processes contributing $Na^+$ not $Cl^-$ to groundwater and changing the ratio of $Na^+/Cl^-$.

One explanation for the excess of $Na^+$ would be that the abundant $Ca^{2+}$ and $Mg^{2+}$ in fresh groundwater exchanges with the $Na^+$ on the surface of clay minerals, which results in an increase of $Na^+$ concentration and a decrease of $Ca^{2+}$ and $Mg^{2+}$ concentration in groundwater (Awaleh et al., 2017). The relationship of $[(Ca^{2+}+Mg^{2+})-(HCO_3^-+SO_4^{2-})]$ vs. $[(Na^++K^+)-Cl^-]$ (Figure 7f) shows a regression line of y=1.0016x+4.9078 ($R^2$=0.9966) and corroborates the contribution of cation exchange ($Ca^{2+}$ or $Mg^{2+}+2NaX$ (solid) $\rightarrow 2Na^++CaX_2$ or $MgX_2$ (solid)) (Verma et al., 2016). In addition, silicate weathering (e.g. $2NaAlSi_3O_8$ (Albite)$+2CO_2+11H_2O \rightarrow 2Na^++Al_2Si_2O_5(OH)_5$ (Kaolinite)$+3H_4SiO_4+2HCO_3^-$) in the aquifers could also

contribute $Na^+$ not $Cl^-$ to groundwater (Guo et al., 2015). The ratio of $Na^+/(Cl^-+SO_4^{2-})$ is around 1 (Figure 7b), demonstrating mirabilite ($Na_2SO_4 \cdot 10H_2O$) dissolution ($Na_2SO_4 \cdot 10H_2O \rightarrow 2Na^+ + SO_4^{2-}$) is an additional strong possible process that could also be responsible for the excess of $Na^+$ compared to $Cl^-$ in groundwater (Jia et al., 2017). Groundwater in the basin with extremely high TDS concentration (more than 1000,000 mg/L) has very low ratios of $Na^+/Cl^-$ (0.08-0.26) as the result of

suspected reverse cation exchange ($Na^+ + CaX_2$ or $MgX_2$ (solid) $\rightarrow 2Na^+ + CaX_2$ or $MgX_2$ (solid)) (Figure 7f).

The relationship between ($Ca^{2+}+Mg^{2+}$) and ($HCO_3^-+SO_4^{2-}$) shows that almost all groundwater from the piedmont to the middle-lower stream area (Zone 2, 3, 4) are plotted along the 1:1 line (Figure 7c), implying the dissolution of minerals such as gypsum, anhydrite, aragonite, calcite and dolomite are the potential ion sources to groundwater in the mineralization process (Dogramaci et al., 2012). As shown in Figure 7d, nearly all groundwater data plot away from the equiline of

($Ca^{2+}+Mg^{2+}$) vs. $HCO_3^-$ (only 3 samples with the ($Ca^{2+}+Mg^{2+}$)/ $HCO_3^-$ ratio in the range of 0.8-1.2, 6 samples with the ratio range of 0.2-0.5, and others with the ratio range of 1.2-604.2), indicating that the $Ca^{2+}$, $Mg^{2+}$ and $HCO_3^-$ are not primarily derived from the dissolution of aragonite, calcite and dolomite. The saturation index values of aragonite, calcite and dolomite are all almost greater than 0 in all samples (Table 4), suggesting the dissolution of these three minerals must be minimal. While the saturation index values of gypsum and anhydrite for groundwater in these areas are all below zero (Table 4),

corroborating the contribution of gypsum and anhydrite dissolution for groundwater mineralization. The deficiency of $Ca^{2+}$ compared to $SO_4^{2-}$ of groundwater (73.5% of samples with the $Ca/SO_4$ ratio less than 0.8) presented in Figure 7e is most likely as a result of the aforementioned mirabilite ($Na_2SO_4 \cdot 10H_2O$) dissolution and cation exchange. As mentioned earlier, the redox conditions of the deep confined aquifers in Zone 4 has evolved to a reduced environment, but due to the extremely low organic carbon content in the sediments (Bowler et al., 1986; Chen and Bowler, 1986), sulfate reduction has a very

limited influence on groundwater chemical evolution. This is also the reason that groundwater in the downstream area (Zone 4 and Zone 5) has an abundant content of $SO_4^{2-}$ in contrast to $Ca^{2+}$.

Groundwater in the low-lying depression (Zone 5) has extremely high TDS values (>300,000 mg/L) (Table 1) and almost all minerals are over-saturation (SI>0) (Table 4), therefore, precipitation (crystallization) of minerals is the primary geochemical process in this part of the aquifers (Li et al., 2010). In addition, reverse cation exchange interaction and evaporation, which

can be confirmed by the relationship of [($Ca^{2+}+Mg^{2+}$)-($HCO_3^-+SO_4^{2-}$)] vs. [($Na^++K^+$)-$Cl^-$] (Figure 7f) and the relation of stable water isotopes (Figure 4), respectively, are also important mechanisms governing groundwater chemistry. Surface water has significant influences on the geochemical processes that occurred in the shallow aquifers. In the wet season, a large amount of relative fresh water can reach the low-lying depression area (Zone 5) and infiltrate to the shallow aquifers. This would dilute the groundwater and dissolve the evaporate salts in the aquifers.

**5.3 Groundwater flow and hydrogeochemiscal evolution**

Theoretically, three types of groundwater flow systems, namely local, intermediate and regional, may occur in a large basin, and each flow system has its own characteristics based on aspects of flow path, recharge origin, cycle depth, cycle amount, residence time, discharge position, hydrochemistry and controlling mechanisms (Toth, 1963). The cross-sectional

groundwater flow modelling results demonstrated the groundwater flow paths in the study area are strictly controlled by distribution of the lithology (Figure 8). Groundwater flow lines are shown to be upward convex in shape at the front of the alluvial fan and the middle-stream area due to an increase in relatively poor permeability due to the addition of finer less permeable stratigraphic material. Based on the distribution of flow lines, three groundwater flow systems including local, intermediate and regional system were identified in the study area (Figure 8).

The local groundwater flow system occurs in the shallow part of the alluvial fan (Zone 2) and overflow zone (Zone 3) with the deepest cycle depth within 250 m of surface. This flow system obtains recharge water along the Golmud River flow path and discharges at the overflow zone. The water cycle quantity of the local system estimated by modelling accounts for approximately 83% of the total quantity of groundwater in the watershed. Groundwater chemistry is mainly controlled by water-rock interaction and there appears to be very little evaporation. Groundwater has a rapid velocity in this part of the system with residence time less than 60 years. As a result, groundwater here is fresh with TDS values less than 1000 mg/L and the water type is mainly $HCO_3 \cdot Cl \cdot Ca \cdot Mg \cdot Na$. This system is the main source of water supply for Golmud city.

The intermediate flow system occurs below the local system and is recharged by river water seepage near the upper part of the alluvial fan. Groundwater flows to lower elevations towards the north and reaches its deepest cycle depth near 600 m at the middle part of the alluvial fan (Zone 2). Due to the increase of aquitards, water flow lines are presented as upward convex shapes at the middle-lower part of alluvial fan (Zone 2). Groundwater flow is constrained by two continuous aquitards (clay layers) at depths of 60 m and 290 m (Figure 8), respectively, at the front of the alluvial fan and overflow zone. The intermediate flow system discharges between the lower overflow zone (Zone 3) and the middle-lower stream area (Zone 4) as evidenced by springs and surface evaporation. The total cycle water quantity of the intermediate system accounts for approximately 14% of the total cycle groundwater amount in the watershed. Aquifers of this system in the alluvial fan have higher renewal rates due to their increased permeability, compared to those in the lower overflow zone (Zone 3) and the middle-lower stream area (Zone 4) that have relative low renewal rates as the result of a lithology dominated by finer sediments, with groundwater residence times of about 4,000 years. Hydrochemistry is dominantly controlled by water-rock interaction, and also strongly influenced by evaporation within the discharge area. Because of the short residence time and a shortage of chemical solutes in the aquifer material, groundwater in the alluvial fan (Zone 2) generally maintains recharge water chemical characteristics which are fresh and $HCO_3 \cdot Cl \cdot Ca \cdot Mg \cdot Na$ type. Sufficient solutes in the aquifer medium and intensive evaporation in the fine soil plain results in the groundwaters gradually evolving to be brackish water and in some cases saline waters.

The regional groundwater flow system occurs under the intermediate system and is recharged at the upper part of the alluvial fan by river water seepage and lateral flow within the mountainous area and discharges at the basin center into terminal lakes resulting in evaporation. Groundwater flow paths are significantly controlled by the lithology (Figure 8), and divided from the intermediate system by a continuous aquitard at a depth of 290 m. Aquifers of this system have very low water renewal rates with residence times up to and greater than 20,000 years. The modelled cycle water quantity of the regional system is only approximately 3%. Groundwater chemistry is mainly influenced by water-rock interaction in this system, except for

shallow aquifers in the discharge area (Zone 5) which are strongly influenced by evaporation. Due to the substantial difference of residence time, water-rock interaction results in much different hydrochemical characteristics from the other aquifer systems. Groundwater that was presented as fresh water with a dominant water type of $HCO_3 \cdot Cl\text{-}Ca \cdot Mg \cdot Na$ in the alluvial fan (Zone 2) and overflow zone (Zone 3), become a brackish water type ($HCO_3 \cdot Cl\text{-}Na$) and a saline water type (Cl-Na) in the middle-lower stream area (Zone 4), and has evolved to be a brine water type mainly composed of Cl-Mg in the low-lying discharge area (Zone 5).

## 6 Conclusions

Previous studies on arid closed basins such as the Great Artesian Basin, Murray Basin, Death Valley and Minqin Basin have established a lot of typical groundwater circulation and evolution regimes. While the Qaidam basin, a typical arid sedimentary closed basin formed with the uplift of the Tibetan plateau, has groundwater circulation patterns characterized by the complex tectonic activities, paleo climate variation, arid climate characteristics, sedimentary lithology, and systematic evolution from fresh to salt water. Studies of this basin can enhance the understanding of groundwater origin, flow regime and hydrogeochemical evolution in such complex tectonic influenced arid sedimentary closed basins worldwide. Integration of hydrogeochemistry, isotopes and 2-dimentional groundwater flow modelling was used to obtain insight into the hydrogeology in a typical arid endorheic watershed represented by the Qaidam Basin, Tibetan plateau. A number of key findings have come out of this study.

The groundwater in the basin originates from precipitation and melt water in the mountainous areas to the south. Groundwater in the alluvial fan is recharged directly as a result of modern river water seepage and mountainous lateral inflow, and has a rapid flow rate. Shallow phreatic waters in the overflow zone and the middle-lower stream area are supported by local and intermediate groundwater flow systems, and have a close chemical and isotopic relationship with surface water. Deep confined groundwater in the overflow zone and the middle-lower stream area are recharged from paleo meteoric water during the late Pleistocene and Holocene under a cold climate based on the results for stable water isotopic analyses. Groundwater in the low-lying depression (basin center) is ancient brines which have possibly migrated from the western Qaidam Basin due to the uplift of the western basin in the geological past. Shallow phreatic aquifers in the low-lying depression (basin center) are also seasonally recharged by modern surface water during flooding periods.

Groundwater in the study area evolves from fresh water to brine water along the flow path. The hydrochemistry of groundwater in the alluvial fan is dominantly controlled by mineral dissolution and cation exchange, and occurs as slightly alkaline water with TDS values less than 1000 mg/L and a water type with a composition of $HCO_3 \cdot Cl\text{-}Ca \cdot Mg \cdot Na$. Deep confined groundwater chemistry in the overflow zone and middle-lower stream area is also controlled by mineral dissolution and cation exchange, as a result of longer residence times in the aquifers, and shows a trend evolving from fresh water to brackish water and finally saline water with increasing solute inputs along the flow paths. As well as water-rock interaction, shallow phreatic water is also affected by intensive evaporation, and therefore, these waters can be much saltier than deep

confined water. Groundwater in the low-lying depression (basin center) is brine water and the mineral precipitation coupled with reverse cation exchange are the dominant geochemical processes controlling water chemistry. The impact of evaporation is also one of the important geochemical processes in the shallow phreatic aquifers, which can accelerate evaporate mineral precipitation. Salt dissolution occasionally occurred in the low-lying depression (basin center) during flood periods due to the infiltration of large amounts of fresh surface water.

Three different hierarchical groundwater flow systems were identified using the cross-sectional model. The continuous aquitards at depths of 60 m, 290 m and 450 m have significant constraints on groundwater flow. The local flow system occurs in the shallow part of the alluvial fan and overflow zone and discharges in the overflow zone with the deepest cycle depth within 250 m of surface. The intermediate system occurs below the local system and discharges in the lower overflow zone and middle-lower stream area with the deepest cycle depth reaching 600 m below surface. The regional system was separated from the intermediate system by a continuous aquitard at a depth of 290 m and discharges in the low-lying depression (basin center). Our calculation shows that the discharge water quantity of these three systems accounts for approximately 83%, 14% and 3%, respectively.

This study enhanced the understanding of the origin, flow pattern, hydrochemical evolution and controlling mechanisms of the regional groundwater systems in the Qaidam Basin. These results can provide fundamental information for coping with future issues such as water conflicts, salt lake exploitation and climate warming in the basin, and also provide references for understanding the hydrogeological processes in other similar endorheic watersheds of northwest China and elsewhere in the world.

**Acknowledgements.** This work was supported by the National Key R&D Program of China [2017YFC0406106] and the China Geological Survey [DD20160291]. We appreciate the help of Ge Zhang and Xiangzhi You at Xi'an Center of Geological Survey, China Geological Survey, Zongyu Chen and Qichen Hao at the Institute of Hydrogeology and Environmental Geology, Chinese Academy of Geological Sciences, Xiaomin Gu at Nantong University, Jingxing Liu and Dong Wang at China University of Geosciences (Beijing). We are grateful to Editor Graham Fogg and the two anonymous reviewers whose insightful comments were very helpful in improving the manuscript.

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

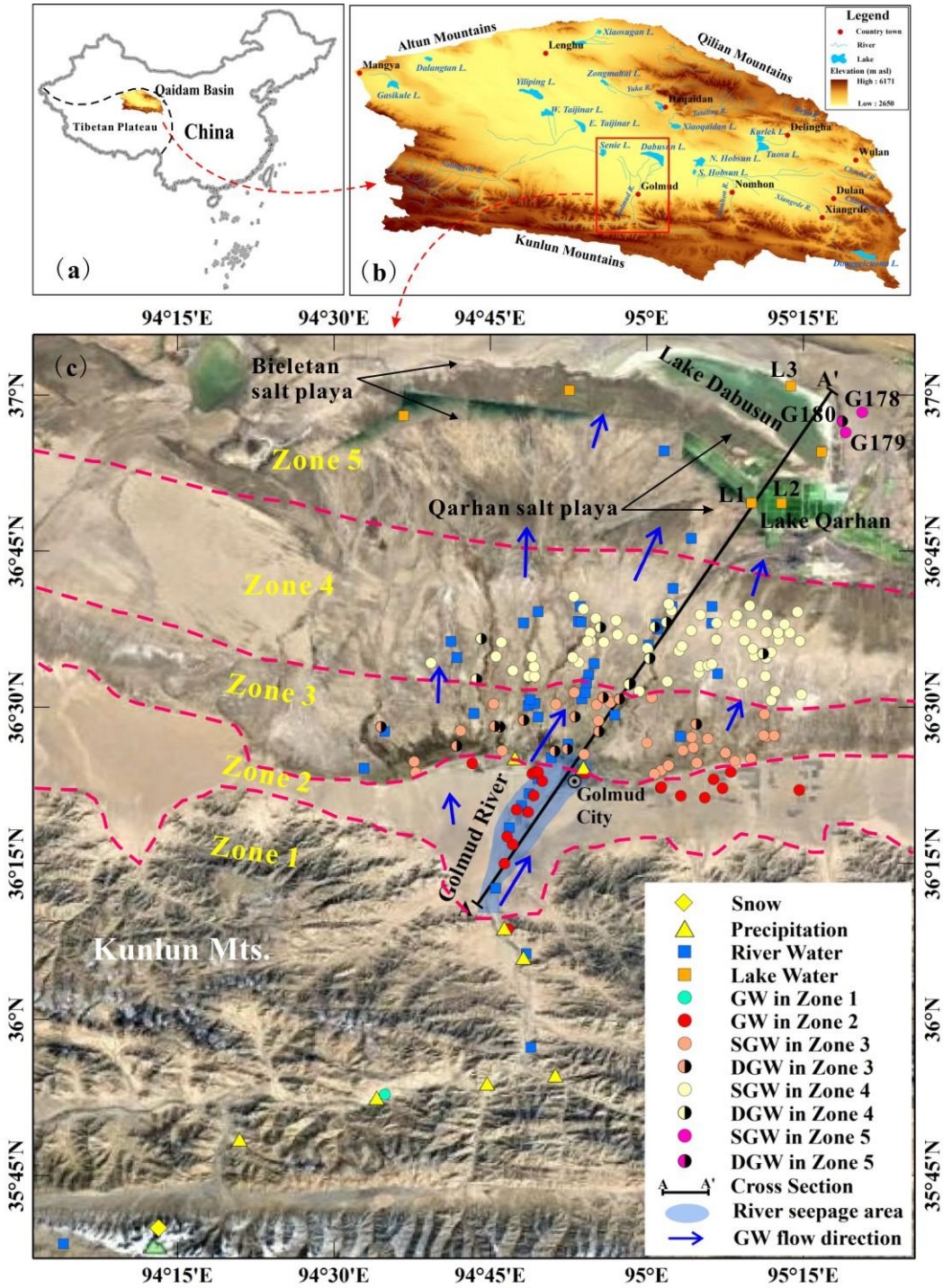

**Figure 1: Location of the study area (a) within China, (b) within the Qaidam Basin, (c) details of sampling location and groundwater/physiographic zones within the study area.**

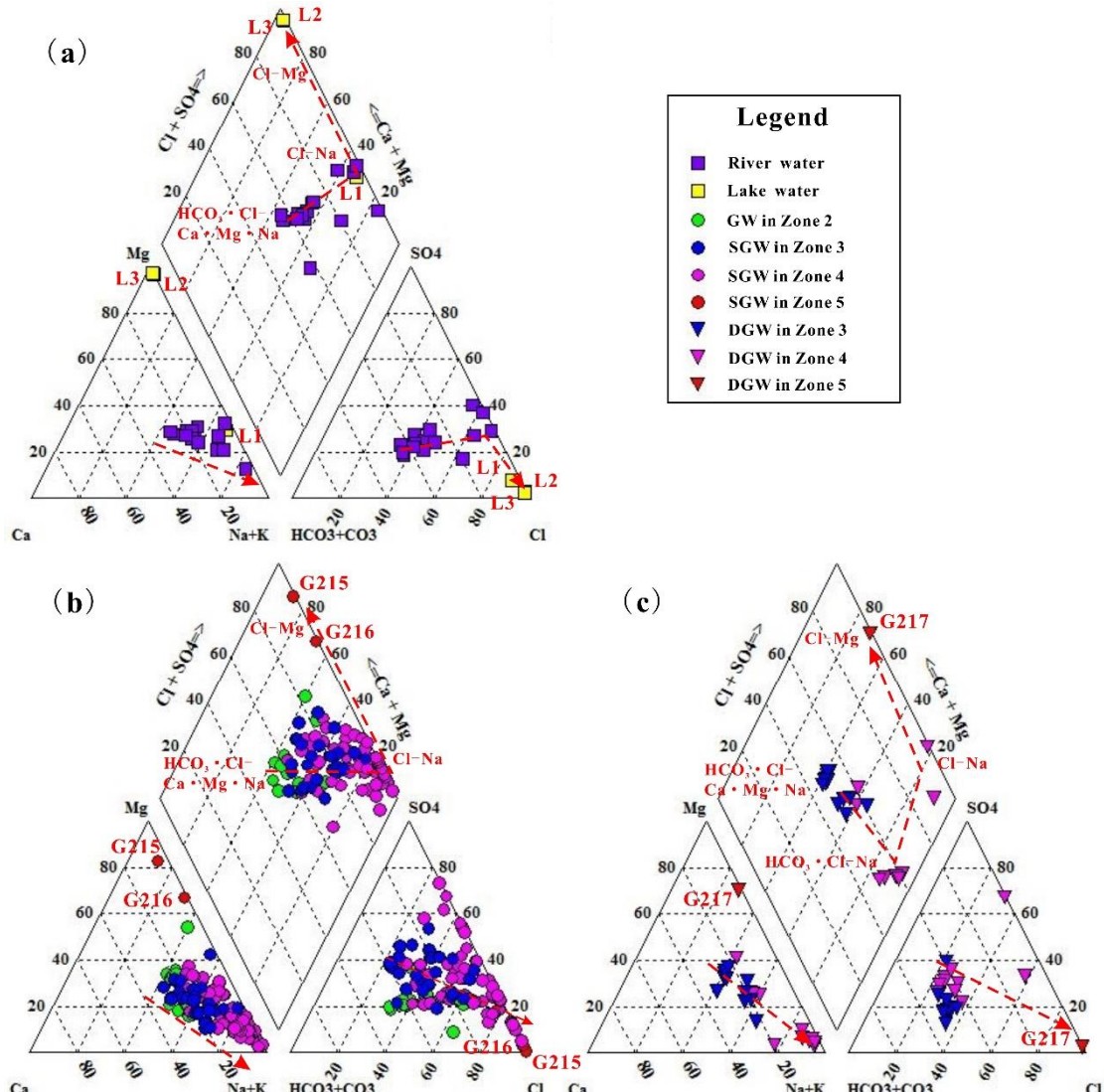

**Figure 2: Piper diagrams of sampled surface water and groundwater. (a) surface waters; (b) shallow phreatic groundwaters; (c) deep confined groundwaters from the Qaidam Basin, China. (Red dashed lines and arrows indicate the direction of evolutionary flow systems).**

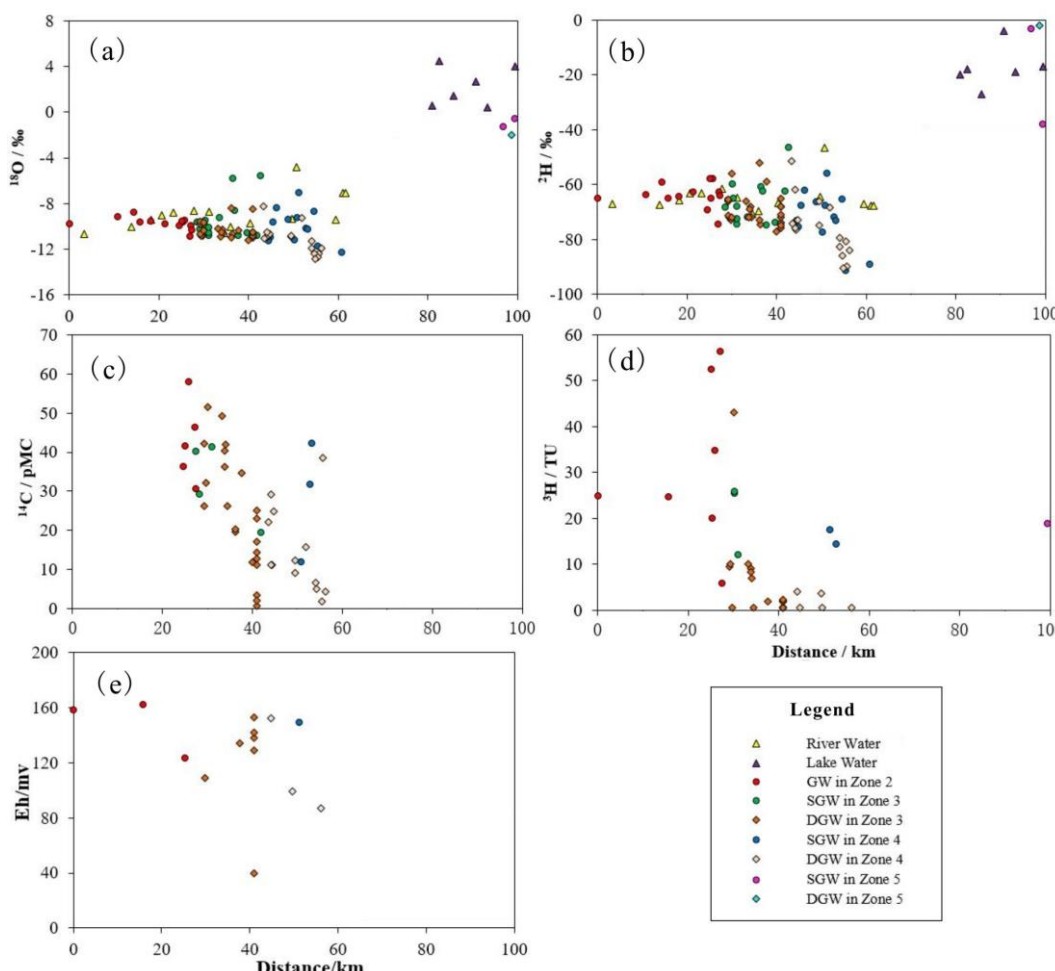

Figure 3: Isotopic data and Eh versus distance from the mountain pass along the groundwater flow paths. (a) $^{18}O$ vs. distance, (b) $^{2}H$ vs. distance, (c) $^{14}C$ vs. distance, (d) $^{3}H$ vs. distance, Eh vs. distance.

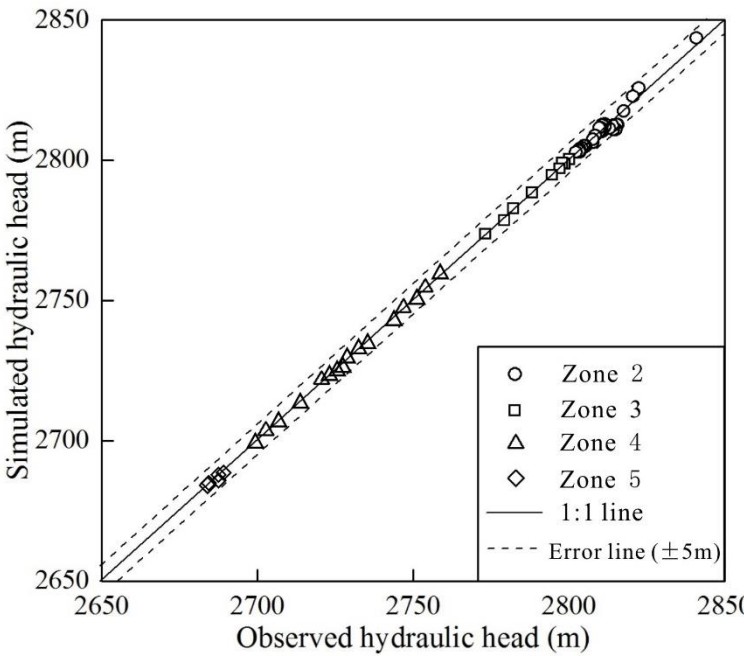

**Figure 4: Comparison of observed and simulated hydraulic head values along the groundwater flow system, Golmud Watershed, China.**

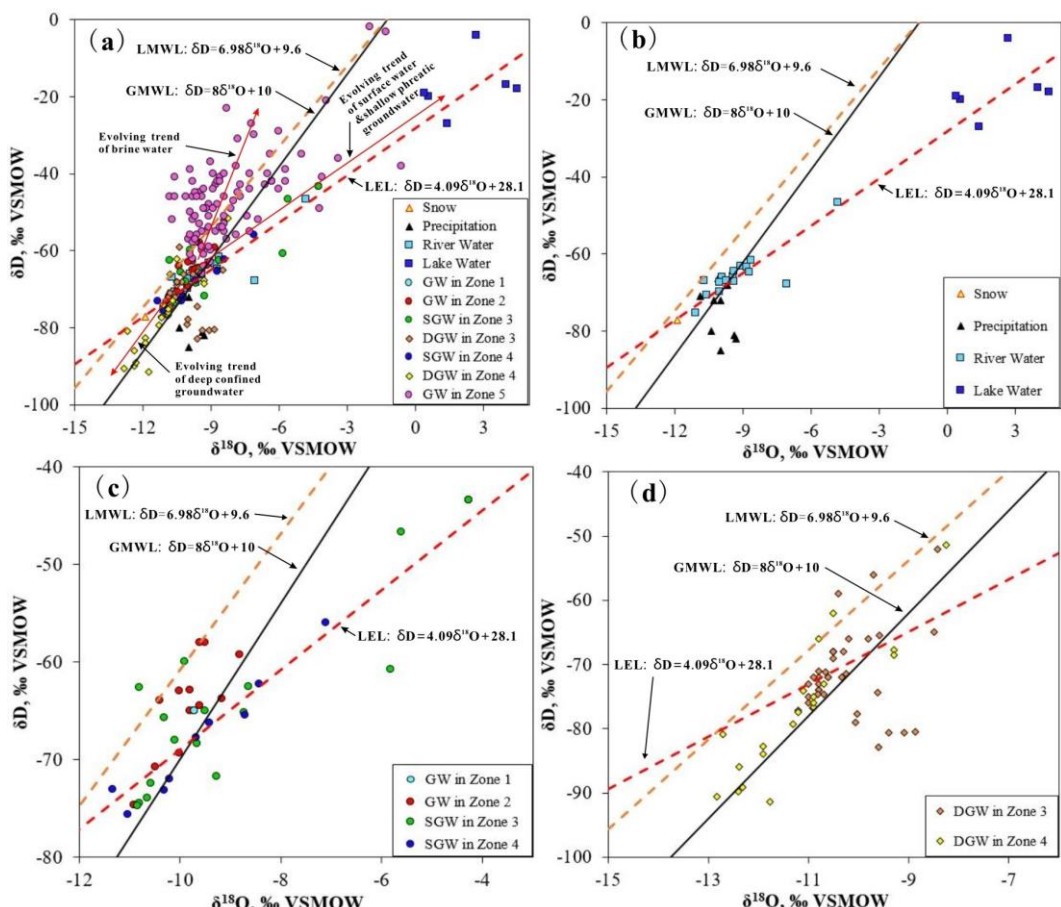

Figure 5: δ¹⁸O vs. δD diagram of precipitation, surface water and groundwater for the Golmud study area of the Qaidam Basin, China. (a) All data, (b) Snow, precipitation and surface waters, (c) Shallow phreatic waters, (d) Deep confined waters.

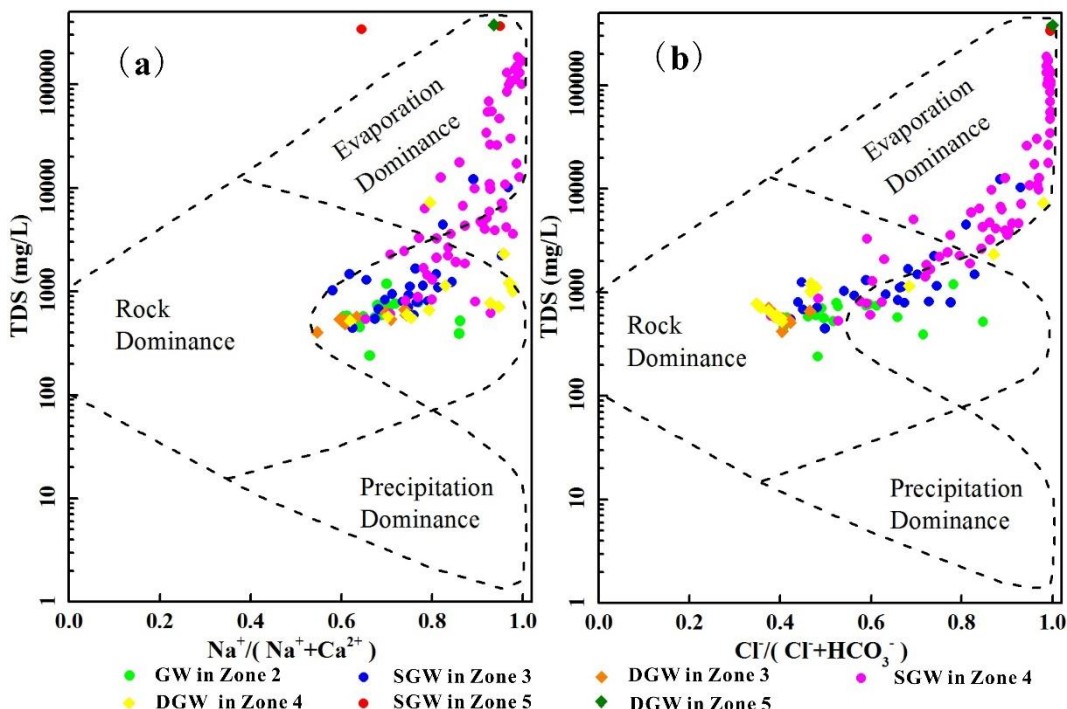

**Figure 6: Diagrammatic representation showing the mechanisms controlling groundwater chemistry. (a) TDS vs. Na$^+$/(Na$^+$+Ca$^{2+}$); (b) TDS vs. Cl$^-$/(Cl$^-$+HCO$_3$$^-$) (after Gibbs, 1970).**

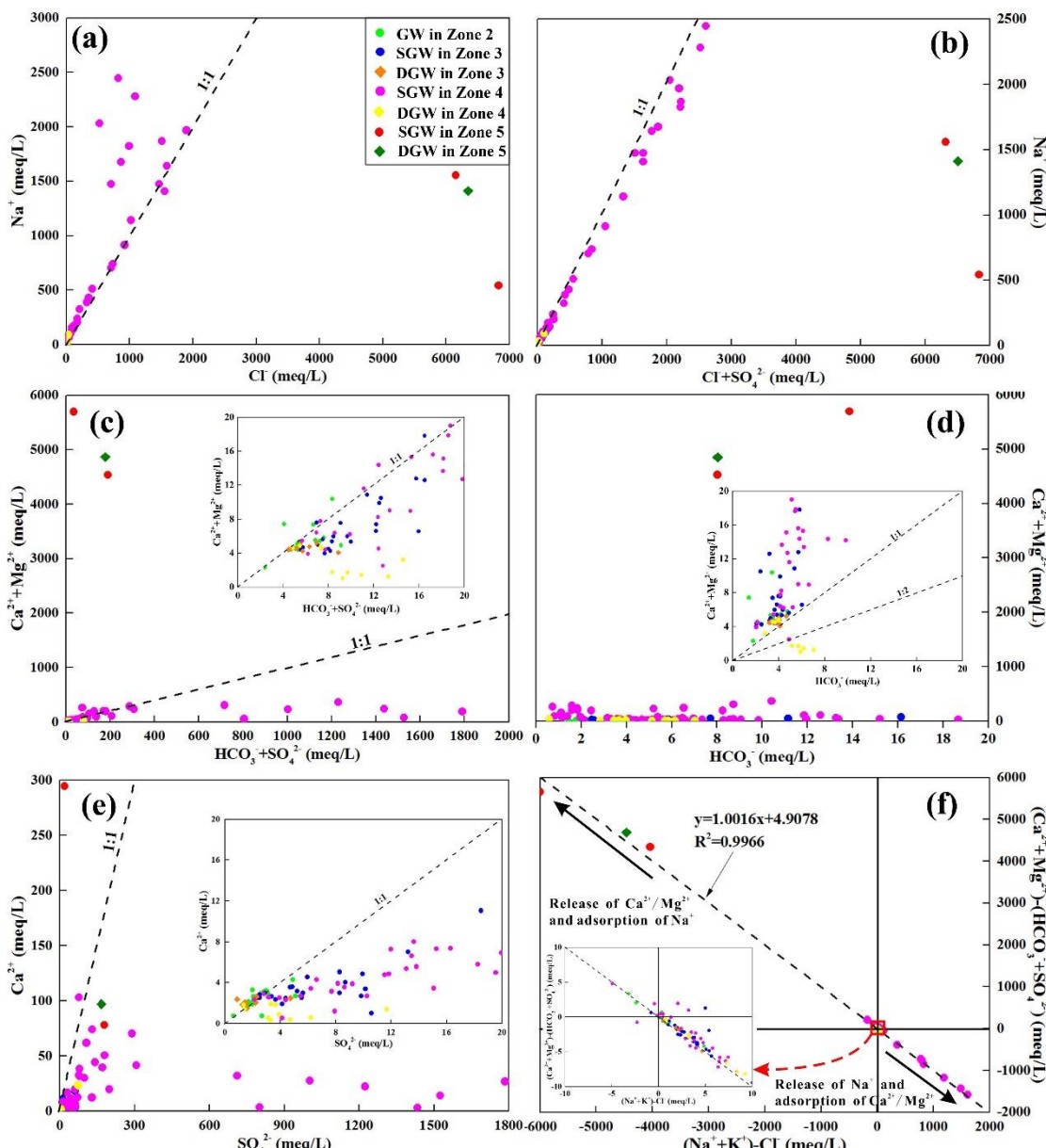

**Figure 7: Bivariate plots (meq/L) of various ions in shallow phreatic and deep confined groundwater showed state (a) Na vs. Cl, (b) Na vs. (Cl+SO₄), (c) (Ca+Mg) vs. (HCO₃+SO₄), (d) (Ca+Mg) vs. HCO₃, (e) Ca vs. SO₄, (f) (Ca+Mg)-(HCO₃+SO₄) vs. [HCO₃+ SO₄].**

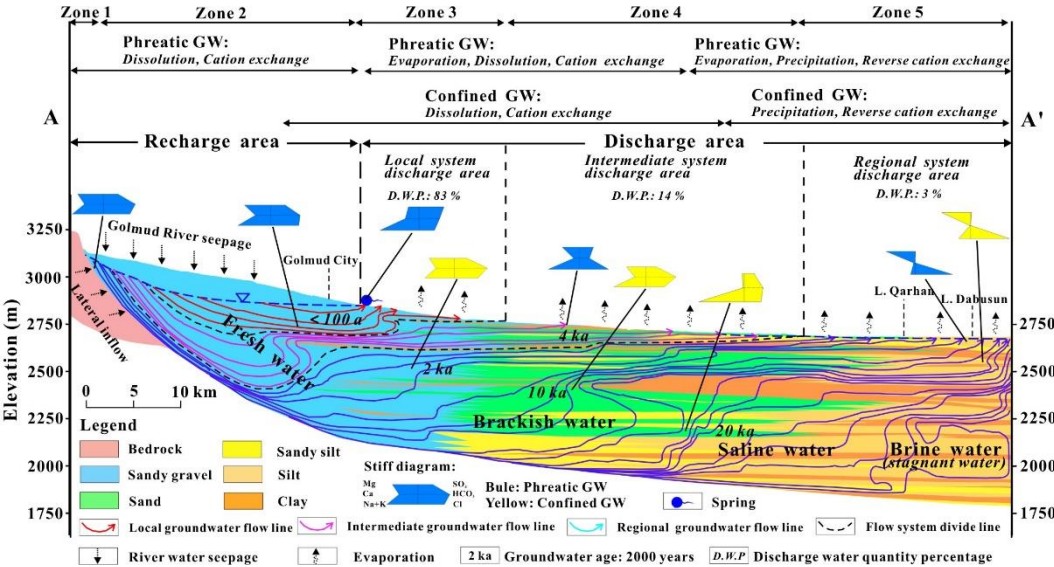

**Figure 8: Conceptual model of groundwater flow and hydrochemical evolution in the Golmud watershed, China.**

**Table 1: Statistical summary of physical and chemical parameters of the surface water and groundwater in the Golmud Watershed, Qaidam Basin, China.**

| Place | Source | | pH | TDS mg/L | Ca mg/L | Mg mg/L | Na mg/L | K mg/L | Cl mg/L | HCO$_3$ mg/L | SO$_4$ mg/L |
|-------|--------|------|------|---------|---------|---------|---------|--------|---------|---------|---------|
| Zone 2 | RW | Min | 8.03 | 393 | 30.9 | 26.3 | 63.0 | 3.7 | 87 | 184.0 | 68.5 |
| | | Max | 8.41 | 523 | 38.9 | 36.4 | 92.1 | 5.3 | 144 | 214.6 | 100.5 |
| | | Mean | 8.28 | 462 | 35.8 | 32.1 | 80.2 | 4.5 | 114 | 198.1 | 85.4 |
| | SGW | Min | 7.62 | 236 | 14.8 | 18.9 | 78.5 | 2.0 | 90.3 | 89.1 | 28.8 |
| | | Max | 8.83 | 1,171 | 86.2 | 80.0 | 232.2 | 10.7 | 436.7 | 309.0 | 235.0 |
| | | Mean | 8.07 | 618 | 49.0 | 36.4 | 131.0 | 6.9 | 173.7 | 221.8 | 116.0 |
| Zone 3 | RW | Min | 8.25 | 368 | 37.7 | 24.4 | 56.1 | 3.5 | 77.4 | 178.1 | 68.7 |
| | | Max | 8.55 | 1,266 | 64.2 | 83.9 | 276.0 | 13.3 | 382.0 | 335.0 | 256.0 |
| | | Mean | 8.43 | 670 | 49.1 | 42.8 | 139.8 | 6.8 | 177.5 | 232.8 | 133.9 |
| | SGW | Min | 7.42 | 443 | 20.6 | 16.9 | 98.0 | 2.0 | 83.1 | 132.0 | 116.2 |
| | | Max | 9.32 | 12,116 | 359.0 | 474 | 3,385.0 | 187.0 | 4316 | 1018 | 3059 |
| | | Mean | 8.09 | 1853 | 81.9 | 81 | 476.9 | 22.7 | 554.9 | 293.0 | 501.9 |
| | DGW | Min | 7.89 | 404 | 30.0 | 18.6 | 65.9 | 4.1 | 84.0 | 204.8 | 41.4 |
| | | Max | 8.64 | 676 | 52.4 | 37.7 | 162.0 | 9.7 | 145.0 | 310.0 | 218.0 |
| | | Mean | 8.19 | 547 | 41.3 | 29.9 | 95.3 | 6.2 | 93.7 | 248.2 | 88.3 |
| Zone 4 | RW | Min | 7.94 | 616 | 39.7 | 38.7 | 104.5 | 5.5 | 152.0 | 246.0 | 97.5 |
| | | Max | 8.64 | 1,833 | 82.6 | 116.9 | 432.2 | 18.7 | 580.7 | 448.0 | 382.3 |
| | | Mean | 8.29 | 1,013 | 55.9 | 57.7 | 232.7 | 9.6 | 289.3 | 303.1 | 191.1 |
| | SGW | Min | 7.08 | 528 | 10.4 | 15.5 | 102.0 | 5.2 | 83.1 | 48.4 | 97.9 |
| | | Max | 9.34 | 185,006 | 2,048 | 4,058 | 56,200 | 1709 | 67,063 | 1179 | 82,202 |
| | | Mean | 8.01 | 32,029 | 322.7 | 635 | 10,464 | 248 | 11,550 | 375 | 8,627 |
| | DGW | Min | 7.83 | 514 | 6.7 | 8.0 | 68.7 | 4.0 | 84.0 | 39.9 | 63.9 |
| | | Max | 8.69 | 7,184 | 456.0 | 48.6 | 2,048 | 39.8 | 1,292 | 445 | 3,648 |
| | | Mean | 8.25 | 1,401 | 59.0 | 24.8 | 406.8 | 8.4 | 263.1 | 269.3 | 483.2 |
| Zone 5 | RW | Min | 8.73 | 1,741 | 56.1 | 92.8 | 437.9 | 12.1 | 655.2 | 159.7 | 208.1 |
| | | Max | 9.45 | 3,268 | 81.0 | 118.5 | 996.3 | 18.9 | 1,776 | 448.3 | 396.8 |
| | | Mean | 9.09 | 2,319 | 70.7 | 106.3 | 628.3 | 15.3 | 1061 | 268.1 | 289.9 |

| | | | | | | | | | | |
|---|---|---|---|---|---|---|---|---|---|---|
| FLW | Representative | 8.98 | 10,937 | 113.7 | 696.3 | 2,957 | 231.8 | 5,912 | 314.9 | 660.7 |
| SLW | Min | 6.03 | 399,098 | 116.2 | 99,500 | 4,137 | 3,168 | 276,849 | 1,941 | 7,717 |
| | Max | 6.28 | 403,758 | 177.2 | 100,240 | 4,740 | 4,122 | 285,780 | 3,118 | 10,894 |
| | Mean | 6.16 | 401,428 | 146.7 | 99,870 | 4,439 | 3,645 | 281,315 | 2,530 | 9,306 |
| SGW | Min | 6.03 | 336,229 | 1,541 | 53,480 | 12,388 | 11,798 | 215,561 | 506.0 | 984.5 |
| | Max | 8.56 | 361,200 | 5,871 | 64,860 | 35,712 | 22,351 | 239,340 | 874.3 | 8,313 |
| | Mean | 7.30 | 348,715 | 3,706 | 59,170 | 24,050 | 17,075 | 227,451 | 690 | 4,649 |
| DGW | Representative | 8.64 | 370,940 | 1,927 | 57,079 | 32,378 | 19,372 | 222,404 | | 7,851 |

Note: RW-River water; SGW-Shallow phreatic groundwater; DGW-Deep confined groundwater; FLW-Relative fresh lake water; SLW-Salt lake water.

5  **Table 2: Statistical summary of isotopic analysis results of precipitation, surface water and groundwater in the Golmud Watershed, Qaidam Basin, China.**

| Place | Source | $\delta D$ ‰ VSMOW | | | $\delta^{18}O$ ‰ VSMOW | | | $^3H$ TU | | | $^{14}C$ pMC | | | $\delta^{13}C$ ‰ | | |
|---|---|---|---|---|---|---|---|---|---|---|---|---|---|---|---|---|
| | | Min | Max | Mean | Min | Max | Mean | Min | Max | Mean | Min | Max | Mean | Min | Max | Mean |
| Zone 1 | SNW | | | -77.0 | | | -11.9 | | | | | | | | | |
| | PW | -85.3 | -71.6 | -75.2 | -10.9 | -9.3 | -10.0 | | | | | | | | | |
| | RW | -75.4 | -64.8 | -68.7 | -11.1 | -9.3 | -10.1 | | | | | | | | | |
| | GW | | | -65.0[a] | | | -9.7[a] | | | | | | | | | |
| Zone 2 | PW | -68.1 | -66.2 | -67.2 | -10.1 | -9.7 | -9.9 | | | | | | | | | |
| | RW | -67.5 | -63.2 | -65.4 | -10.7 | -8.8 | -9.6 | | | | | | | | | |
| | SGW | -74.7 | -58.0 | -64.5 | -10.9 | -8.8 | -9.8 | 20.0 | 56.3 | 35.5 | 30.6 | 57.9 | 42.5 | | | |
| Zone 3 | RW | -70.8 | -46.7 | -63.6 | -10.6 | -4.8 | -8.8 | | | | | | | | | |
| | SGW | -74.8 | -43.4 | -63.5 | -10.8 | -4.3 | -8.8 | 12.1 | 25.7 | 21.1 | 19.5 | 41.2 | 32.5 | | | |
| | DGW | -82.9 | -52.1 | -71.6 | -11.2 | -8.4 | -10.3 | <1 | 10.1 | | 0.7 | 49.2 | 23.4 | -4.6 | -1.7 | -2.8 |
| Zone 4 | RW | -67.7 | -67.2 | -67.5 | -9.4 | -7.1 | -8.2 | | | | | | | | | |
| | SGW | -75.6 | -56.0 | -68.0 | -11.3 | -7.1 | -9.6 | 14.4 | 17.5 | 16.0 | 11.9 | 42.2 | 27.3 | | | |
| | DGW | -91.3 | -51.4 | -77.1 | -12.8 | -8.3 | -11.2 | <1 | 4.1 | | 1.9 | 38.6 | 14.8 | -5.5 | -3.9 | -4.8 |
| Zone 5 | RW | | | -51.3[a] | | | -6.6[a] | | | | | | | | | |

| | | | -20.0 [a] | | | 0.6 [a] | |
|---|---|---|---|---|---|---|---|
| FLW | | | -20.0 [a] | | | 0.6 [a] | |
| SLW | -27.0 | -4.0 | -17.0 | 0.4 | 4.5 | 2.6 | |
| GW | -66.0 | -2.0 | -46.3 | -10.8 | -0.6 | -8.2 | 18.9 [a,b] |

Note: SNW-Snowmelt water; PW-Precipitation water; RW-River water; SGW-Shallow phreatic groundwater; DGW-Deep confined groundwater; FLW-Relative fresh lake water; SLW-Salt lake water; a-Only one representative sample data; b-Shallow phreatic water sample data.

**Table 3: Estimated parameters of different lithology from the Golmud Watershed, Qaidam Basin, China.**

| Lithology | $K_h$ (m/d) | Anisotropy ratio $K_h/K_v$ | Porosity |
|---|---|---|---|
| Gravel sand | 56.3 | 10 | 0.35 |
| Sand | 13.7 | 10 | 0.40 |
| Sandy silt | 0.62 | 5 | 0.5 |
| Silt | 0.13 | 5 | 0.6 |
| Clay | 0.001 | 5 | 0.65 |

**Table 4: Saturation index of selected minerals from the Golmud Watershed, Qaidam Basin, China.**

| Place | | Halite | Gypsum | Anhydrite | Aragonite | Calcite | Dolomite | Sylvite |
|---|---|---|---|---|---|---|---|---|
| | Min | -6.87 | -3.27 | -3.75 | -0.08 | 0.08 | 0.45 | -7.59 |
| SGW of Zone 2 | Max | -5.72 | -1.88 | -2.36 | 2.03 | 2.18 | 4.29 | -6.90 |
| | Mean | -6.43 | -2.55 | -3.02 | 1.29 | 1.45 | 2.85 | -7.25 |
| | Min | -6.70 | -3.12 | -3.60 | -0.16 | -0.01 | -0.78 | -7.82 |
| SGW of Zone 3 | Max | -3.70 | -0.90 | -1.36 | 2.24 | 2.40 | 4.97 | -4.41 |
| | Mean | -5.84 | -2.04 | -2.52 | 1.07 | 1.23 | 2.25 | -6.66 |
| | Min | -6.98 | -3.06 | -3.52 | -0.42 | -0.26 | -1.35 | -7.65 |
| DGW of Zone 3 | Max | -6.40 | -2.67 | -3.13 | 1.73 | 1.88 | 3.98 | -7.07 |
| | Mean | -6.82 | -2.87 | -3.33 | 0.89 | 1.05 | 1.91 | -7.46 |
| SGW of Zone 4 | Min | -6.67 | -3.18 | -3.64 | -0.89 | -0.74 | -2.00 | -7.54 |

|  |  |  |  |  |  |  |  |  |
|---|---|---|---|---|---|---|---|---|
|  | Max | -1.16 | 0.32 | -0.06 | 2.43 | 2.58 | 5.65 | -2.44 |
|  | Mean | -3.93 | -1.27 | -1.71 | 1.47 | 1.63 | 3.45 | -5.00 |
|  | Min | -6.96 | -3.80 | -4.28 | -1.02 | -0.87 | -2.49 | -7.57 |
| DGW of Zone 4 | Max | -4.30 | -0.16 | -0.65 | 1.39 | 1.54 | 3.41 | -5.52 |
|  | Mean | -6.06 | -2.89 | -3.36 | 0.05 | 0.20 | 0.06 | -7.16 |
|  | Min | 0.04 | 0.03 | -0.16 | 2.29 | 2.44 | 6.78 | -0.35 |
| SGW of Zone 5 | Max | 0.34 | 0.31 | 0.11 | 2.82 | 2.98 | 7.36 | -0.17 |
|  | Mean | 0.19 | 0.17 | -0.03 | 2.56 | 2.71 | 7.07 | -0.26 |
| DGW of Zone 5 |  | 0.39 | 0.34 | 0.14 | 2.18 | 2.34 | 6.64 | -0.21 |