# Peer review of "Groundwater origin, flow regime and geochemical evolution in arid endorheic watersheds: a case study from the Qaidam Basin, Northwest China"

_Hydrology and Earth System Sciences, 2017_

## Referee Comment (RC1) · Anonymous Referee #1 · 20 Dec 2017

Review of Xiao, Y., and others paper, (in press paper) hess-2017-647 Submitted on 02 Nov 2017 Groundwater origin, flow regime and geochemical evolution in arid endorheic watersheds: a case study from the Qaidam Basin, Northwest China Yong Xiao, Jingli Shao, Shaun K. Frape, Yali Cui, Xueya Dang, Shengbin Wang, and Yonghong Ji

1. Interesting paper that should be published, but authors should respond to a few comments. a. Agree with the argument that there may be three different groundwater flow systems as evidenced by the numerical model and the increased age and lighter isotopic values downdip in the flow system. b. Authors should have run their chemical

analyses through a geochemical equilibrium program to determine degree of mineral saturation. They comment on the fact that there are significant rock water reactions, which is correct, but they should have provided some further documentation as to what minerals are important and whether the water are at saturation. This would give further credence to their geochemical argument. c. The authors did not mention anything about the redox system, which may be important . Are there any organics in the sediments. Is sulfate reduction occurring down the flow paths, especially when you get to the salt lakes and playas. d. The authors do not provide much documentation that the brines in Zone 5 have migrated into the downdip section of the flow system from some other location during an earlier time period. Presumable evaporates have been accumulated at this location since the Pleistocene, and the chemistry observed results from in situ rock/water interactions, and not the migration in from other location. e. Figure 6 and 7 may be too complicated. Data for different flow systems might be better represented as individual graphs. f. How does intense evaporation occur at the water table (i.e. a few meters below land surface)? It may well well be out of an evaporation zone. Are caliches developing? g. Paper should be considered as a reconnaissance level paper, opening the door for the authors to look at their conclusions in greater detail and greater analysis.

---

## Referee Comment (RC2) · Anonymous Referee #2 · 12 Mar 2018

General comments

This is potentially and interesting study and it is a nice dataset. However, it needs to have much more work done on it before it is publishable in a major journal such as HESS. The paper is brief and there is a lack of justification of key points. In particular, the interpretation of the 14C and the major ions is speculative at best. There is also insufficient details on the hydrogeological framework to understand the data in context.

I hope that these comments are useful in revising the paper.

[Figure]

Abstract

The Abstract provides a reasonable summary of the paper, some specific comments.

1) Line 13 – be specific about which results show this.

2) The water types (lines 18-20) are just a descriptive and do not by themselves indicate much about process. "Water-rock interaction" and "evaporation-precipitation" also are general descriptions. It would be better to specify exactly what minerals are involved in these reactions.

3) The % of water from the systems (line 22) cannot be that precisely estimated.

Introduction

The introduction references a lot of literature, but needs more details. You should outline the specific issues that you are addressing in this paper more fully – it is good to mention a range of features (resources, ecology etc) but the main purpose of this paper, which is to understand the hydrogeology, needs more emphasis. You should expand this section to explain in more detail how this work specifically addresses an important hydrogeology question and how it relates to our understanding of groundwater in these types of basins in general.

HESS is an international journal and so papers need to appeal to readers working in other regions, so it is critical that you explain the general importance of the work. Perhaps refer to basins elsewhere and explain the common questions that this study will help to address.

Specific comments

Page 2 lines 13-17. This just says that it was difficult to do the research; perhaps more importantly is some indication as to why this information would be useful.

Page 2 lines 18-27. Be specific with the term "isotopes" as there is a considerable difference between the information that you get from the stable isotopes (O & H) and

radioactive isotopes such as 14C. Better to specify which ones.

You need to develop the aims better. You can do this either by framing a hypothesis or by explaining the aims more fully. At the moment, you just say that there are some techniques that we can use to help us understand groundwater systems and you are going to apply them to this basin. What specifically do you hope to achieve and how will it inform the understanding of this basin and similar ones?

Study Area

The study area section needs more detail. This is a hydrochemistry paper that as background requires an understanding of the hydrogeology. However, many details are lacking, such as:

1) You should describe what is known about the flow system, for example where are the recharge and discharge areas?

2) The maps should show recharge areas and groundwater flow paths

3) What do we know about hydraulic properties (especially K)?

Without this information it is very difficult to understand the study and the statement "Overall, groundwater in the basin originates from Golmud River seepage and bedrock lateral flow in the alluvial fan, and topography results in flow towards the low-lying depression (basin center)." is hard to assess.

Materials and Methods

For the groundwater samples, you need more details on the wells. The interpretation of data from long-screened production wells is more difficult than from monitoring bores with short screens and no pumping.

I do not see a data table in the paper (the tables are just summaries). It is critical that you provide the raw data (HESS will let you do this as a supplement). For the groundwater samples, you need to specify

a) The well depths and typical screen intervals

b) Whether these are monitoring or production wells

c) The aquifer that is sampled

The table also needs your geochemical data in it and details such as sampling date etc. The actual data is an important part of this study and must be made available with the paper.

The 14C analyses involved using a field precipitation technique. As discussed by Aggarwal et al. (2014: Groundwater, 52, 20-24) this is prone to errors by atmospheric contamination. Did you assess the possibilities that atmospheric contamination has occurred (using field blanks or repeated samples)? At the very least, you should discuss this.

Page 4, line 22. It is not clear what you mean by "The standard deviation of analytical results ranges between 0.7 pMC and 1.0 pMC" Is this from repeat analyses or is it a typical range from the lab (and why the range of values?).

As noted above, you need to include these data in a Table

Section 3.2. There are many values in here but little indication as to where they come from. Some of these details need to be in Section 2 as they are part of the background understanding. Without a clear description of the hydrogeology and flowpaths it is difficult to assess the appropriateness of the modelling (recharge is mentioned here, but it has not been explained where the recharge areas are and what any prior estimates of recharge are).

Section 4

Table 1 is only a summary table, we need the data!

This section is a reasonable description of the data. However, it is a little brief in places and as discussed below, this does not help with the interpretation. I am not sure that

defining water types is that useful as ion ratios are probably more use for understanding processes (it is a common but slightly outdated way of discussing hydrogeochemistry). While it is true that waters do "evolve" in composition from rainfall to brines (page 6, lines 13-20), that tells us little about the processes that cause them to do so.

There is some material here that is discussion and so belongs later (eg page 6, lines 9-10) and some overly speculative conclusions (eg page 6, lines 19-20) that need to be properly justified or omitted. Conversely, there is some material in section 5 (such as the description of the Na/Cl ratios) that should be here as this is where you present the data.

This section needs restructuring so that all the data that you use to make interpretations is presented and described adequately.

Section 5 Perhaps due to the tendency to explain aspects of the study briefly, and a lack of primary data, there are several conclusions in this section that are questionable (or at least need more explanation).

The interpretation of the 14C residence times (page 9) has several issues:

a) Most importantly, you seen to have samples with low 14C but measurable Tritium (Fig. 3). If so these must be mixtures between older water (low 14C, 3H free) and younger water (high 14C and high 3H) as the time required for measurable 14C to decay wipes out all the 3H. You CANNOT calculate residence times from such waters. A clear explanation of how the radioisotopes are behaving is required before you do any calculations

b) Secondly, just saying that you applied the Tamers model or an unspecified statistical is inadequate. The correction of 14C ages is commonly difficult and needs more justification (there are numerous papers that address this in many basins). Even if in the end you just use a simple correction, you need to justify that you understand what is happening in the C-system and rule out possibilities such as open-system carbonate

dissolution or methanogenesis. Also you need to outline what the % of dead carbon is and whether that is reasonable.

c) The Tamers model implicitly assumes that only carbonate dissolution occurs. However, in moderately saline groundwater, you may have carbonate precipitation. Have you assessed this?

d) Do you have 13C data? They would help in the correction process.

e) If you are going to calculate ages, you need to discuss them formally (what is the range, what are the uncertainties etc?)

The interpretation of the evaporation based on the stable isotopes (page 8, lines 15-17) seems to imply that evaporation is occurring along the flow paths. However, evaporation is something that can only occur at or near the surface and not directly from deeper groundwater. Do you mean that different degrees of evaporation occurs in different areas during recharge? If so, how does that fit with your conceptualisation of the hydrogeology?

In several places, the similarity or differences in the geochemistry are mentioned, but there are no attempts to quantify this (the reader has to basically look at the figures or tables and make their own assessment). At the very least put the ranges and differences in means in the text, but preferably try to use something like PCA or ANOVA to better justify this.

Section 5.2 on the hydrogeochemistry also has several issues.

a) The first part of the section is really just an (old) textbook introduction. Yes, these processes control the geochemistry, but the details are more subtle than this. Figure 6 is just a broad generalisation and while it is a useful conceptualisation, it does not tell us much about specific processes (which is the objective here)

b) The subsequent statements on lines 22-30 are unjustified. You need to relate this discussion back to the description of the geochemistry and explain specifically how

you came to these conclusions (ie what in the hydrogeochemistry tells us that water-rock interaction has occurred, what minerals are dissolving / precipitating etc). At the moment your interpretation just relies on where in the system the water is from (this might be correct but it does not make use of any of the geochemistry).

c) Na/Cl ratios are nor definitive in constraining halite dissolution. Rainfall has Na/Cl ratios that are close to 1 (generally 0.7-1.2) and given that ion exchange may also occur, you cannot distinguish evaporation and halite dissolution. Really you need Br and to look at Cl/Br ratios as halite dissolution produces Cl/Br ratios that are orders of magnitude higher than halite dissolution. The SI indices are not relevant.

d) The explanation of the geochemical processes on page 10 would be helped if the mineralogy of the aquifer had been properly described (which of any of these minerals exist – that is obviously important).

e) Finally, it is not clear why the authors have looked at the geochemistry in this much detail. While understanding the geochemistry is important, it should inform a broader understanding of the system, for example: does it constrain inter-aquifer mixing or where the water was recharged, is it useful for interpretation of 14C ages, is there a palaeowater signal in the major ions as well in the stable isotopes that could be useful in detecting climate influences elsewhere. As it is, this section stands alone and is actually not well integrated to the study.

Conclusions

This is a reasonable summary of the main findings of the paper. However, as with the introduction, it needs a couple of extra paragraphs to explain the relevance to researchers working elsewhere (what perhaps have you done differently / better to other studies, are there any general points in understanding basins that you can make?). This will give the paper considerably more impact.

[Figure]

647, 2017.

---

## Author Comment (AC1) · 30 Apr 2018

Response to Referee #1 Dear Referee #1, Thank you very much for the insightful comments concerning our manuscript entitled "Groundwater origin, flow regime and geochemical evolution in arid endorheic watersheds: a case study from the Qaidam Basin, Northwest China" (Manuscript NO.: hess-2017-647). Your comments to manuscript are very valuable and helpful. We have carefully studied and incorporated them into our revised manuscript. Please see the point-to-point response to your comments as following. The revised manuscript is attached in the supplement.

[Figure]

Comment a. Agree with the argument that there may be three different groundwater flow systems as evidenced by the numerical model and the increased age and the lighter isotopic values downdip in the flow system.

Comment b. Authors should have run their chemical analyses through a geochemical equilibrium program to determine degree of mineral saturation. They comment on the fact that there are significant rock water reactions, which is correct, but they should have provided some further documentation as to what minerals are important and whether the water are at saturation. This would give further credence to their geochemical argument.

Reply: We quite agree with this comment. The truth is that we have done this work in the initial submission version, and the results are presented in Table 4. You can find relative discussions in Page 12 lines 21-22, Page 13 lines 13-16, 23-25 as follows. Page 12 lines 21-22: "The calculated results of halite saturation index ($SI_{halite}<0$) (Table 4) confirm that halite minerals of the aquifer matrix could be readily available to the groundwater." Page 13 lines 13-16: "The saturation index values of aragonite, calcite and dolomite are all almost greater than 0 in all samples (Table 4), suggesting the dissolution of these three minerals must be minimal. While the saturation index values of gypsum and anhydrite for groundwater in these areas are all below zero (Table 4), corroborating the contribution of gypsum and anhydrite dissolution for groundwater mineralization." Page 13 lines 23-25: "Groundwater in the low-lying depression (Zone 5) has extremely high TDS values (>300,000 mg/L) (Table 1) and almost all minerals are over-saturation ($SI>0$) (Table 4), therefore, precipitation (crystallization) of minerals is the primary geochemical process in this part of the aquifers (Li et al., 2010)." We are very sorry for that the Table 4 is missing in the Discussion Paper due to some unexpected mis-operation. Now it has been added in the revised manuscript.

Comment c. The authors did not mention anything about the redox system, which may be important. Are there any organics in the sediments? Is sulfate reduction occurring down the flow paths, especially when you get to the salt lakes and playasïij§

Reply: We have added the redox potential info of groundwater in Page 7 lines 7-9 as following. "The redox potential (Eh) of SGW are in the range of 123-162 mV from alluvial fan to middle lower stream area (Zone 2, Zone 3 and Zone 4), suggesting an oxidation condition. The Eh values of DGW vary from 153 mV to 40 mV along the flow path (Zone 3 to Zone 4), indicating the redox condition gradually evolves from oxidation to reduction (Fig. 3e)." It can be known that shallow phreatic groundwater from alluvial fan (Zone 2) to middle lower stream area (Zone 4) is in oxidation condition. While the redox condition of deep confined groundwater from overflow area (Zone 3) to middle lower stream area (Zone 4) evolves from oxidation to reduction. Although we do not have the redox potential data of the basin center (Zone 5), it can be assumed that the redox condition of deep confined aquifers is reduction according to the Eh values of deep confined groundwater in Zone 4. As the sediments in the downstream area have very low organic carbon content, which was reported in the literature (Bowler et al., 1986; Chen and Bowler, 1986), sulfate reduction has very limited influence on groundwater chemical evolution. This discussion has been added on Page 13 lines 18-22 as following: Page 13 lines 18-22: "As mentioned earlier, the redox conditions of the deep confined aquifers in Zone 4 has evolved to a reducted environment, but due to the extremely low organic carbon content in the sediments (Bowler et al., 1986; Chen and Bowler, 1986), sulfate reduction has a very limited influence on groundwater chemical evolution. This is also the reason that groundwater in the downstream area (Zone 4 and Zone 5) has an abundant content of $SO_4^{2-}$ in contrast to $Ca^{2+}$."

Comment d. The authors do not provide much documentation that the brines in Zone 5 have migrated into downdip section of the flow system from some other location during an earlier time period. Presumable evaporates have been accumulated at this location since the Pleistocene, and the chemistry observed results from in situ rock/water interactions, and not the migration in from other location.

Reply: As seen in figure 5, the stable hydrogen and oxygen isotopes of brines in Zone 5 is quite different from the upper stream. Specifically, all groundwaters in Zone 5

are with relative enriched $\delta$D and $\delta$18O values in contrast with modern water and deep confined groundwater in the middle lower stream area (Zone 4), and deviate away from the LEL and towards the left. This may imply their quite different recharge sources or recharge environment. Previous researches reported that the depocenter of Qaidam basin gradually migrated from the northwest to the Dabusun lake area since Late Oligocene (Chen and Bowler 1980; Zhang 1987; Huang 2007). In this depocenter migration process, the paleo lake water and groundwater, both of which were brines, also migrated to the Dabusun lake area. After this, the basin experienced several arid climate cycle, as a result, evaporates precipitated from these paleo migration waters and also the newly recharged waters of this watershed. Briefly, the evidence of brines migration is the tectonic activity and paleo lake migration in the geological history reported in previous literatures. This was briefly stated in Page 11 line 26-27. To provide more evidence, we add two more literatures (Zhang (1987) and Huang (2007)), in the Page 11 lines 26-27. [1] Chen, K., Bowler, J. M., 1986. Late Pleistocene evolution of salt lakes in the Qaidam basin, Qinghai province, China. Palaeogeography Palaeoclimatology Palaeoecology, 54 (1-4):87-104. [2] Zhang, P. X., 1987. Salt lakes in Qaidam Basin. Science Press, Beijing. [3] Huang, L., Han, F. Q., 2007. Evolution of salt lakes and palaeoclimate fluctuation in Qaidam Basin. Science Press, Beijing.

Comment e. Figure 6 and 7 may be too complicated. Data for different flow systems might be better represented as individual graphs.

Reply: We agree that it might be much more clear to present the data for different flow systems in individual graphs. In the current presentation, the data has been separated according to the physiographic zones and phreatic/confined aquifers and represented as different legend, which is enough to illustrate the hydrogeochemiscal evolution. But if we separate the data for different flow systems in individual graphs, there will 4 more pictures (two's size like Figure 6 and two's size like Figure 7) in the manuscript, which will greatly increase the manuscript length. Therefore, we suggest present the data as different legend according to the sampling physiographic zones and phreatic/confined

aquifers like the current presentation.

Comment f. How does intense evaporation occur at the water table (i.e. a few meters below land surface)? It may well well be out of an evaporation zone. Are caliches developing?

Reply: Generally, the evaporation is very intense in the basin due to the hyper-arid climate. But its influence is very limited and can be ignored in the alluvial fan where the groundwater table is tens of meters below the ground surface. For Zone 3~5, the depth of phreatic groundwater is very small, all within 3 meters, as a result, intense evaporation has a significant influence on groundwater chemistry, which can be shown by the stable isotopes presented in Figure 5c and hydrochemistry in Figure 6, as well as the change of minerals saturation states presented in Table 4. Caliches are widely found developing on the ground in the downdip section (including the north part of Zone 4 and the whole Zone 5, see the Picture A.

Comment g. Paper should be considered as a reconnaissance level paper, opening the door for the authors to look at their conclusions in greater detail and greater analysis.

Reply: In this revised manuscript, some analysis (e.g. groundwater age dating) and conclusions have been further expanded and discussed. All of this can make the conclusions and analysis more clear and credible. In addition, some extra sentences are also added on Page 15 lines 9-14 to further illustrate the purpose and importance of this study as follows. Page 15 lines 9-14: "Previous studies on arid closed basins such as the Great Artesian Basin, Murray Basin, Death Valley and Minqin Basin have established a lot of typical groundwater circulation and evolution regimes. While the Qaidam basin, a typical arid sedimentary closed basin formed with the uplift of the Tibetan plateau, has groundwater circulation patterns characterized by the complex tectonic activities, paleo climate variation, arid climate characteristics, sedimentary lithology, and systematic evolution from fresh to salt water. Studies of this basin can enhance the understanding of groundwater origin, flow regime and hydrogeochemical evolution

in such complex tectonic influenced arid sedimentary closed basins worldwide." This can let the readers clearly know what have done in this study and the general points which can provide references for similar basins worldwide. Generally speaking, this study integrates hydrogeochemistry, environmental isotopes and numerical modelling approach based on the current material and data to get reconnaissance insights into the origin, flow pattern and geochemical evolution of regional groundwater from mountain pass to terminal lake in a typical endorheic watershed of Qaidam Basin. The origin and its recharge characteristics of groundwater, especially the brines in the terminal lake area, have been firstly identified in Qaidam Basin based on multiple evidences (e.g. isotopic evidences, tectonic activities in the geological past). Additionally, the regional groundwater flow and geochemical evolution are systematically established in this study. All above can provide a reconnaissance understanding of hydrogeological regimes in the study area and also provide references for similar basins worldwide.

Please also note the supplement to this comment:
https://www.hydrol-earth-syst-sci-discuss.net/hess-2017-647/hess-2017-647-AC1-supplement.zip
* * *
**Fig. 1.** Picture A: The developed caliches in the Zone 4 and 5.

---

## Author Comment (AC2) · 30 Apr 2018

Dear Referee #2, Thank you very much for the insightful comments concerning our manuscript entitled "Groundwater origin, flow regime and geochemical evolution in arid endorheic watersheds: a case study from the Qaidam Basin, Northwest China" (Manuscript NO.: hess-2017-647). Your comments to manuscript are very valuable and helpful. We have carefully studied and incorporated them into our revised manuscript. Please see the point-to-point response to your comments as following. You can find the revised manuscript in the supplyment.

Abstract Part:

Comment 1): Line 13 – be specific about which results show this.

Reply: It has been added as following "The stable isotopes results show ..." (The revised manuscript, Page 1 line 14).

Comment 2): The water types (lines 18-20) are just a descriptive and do not by themselves indicate much about process. "Water-rock interaction" and evaporation-precipitation" also are general descriptions. It would be better to specify exactly what minerals are involved in these reactions.

Reply: The "Water-rock interaction" and evaporation-precipitation" have been specifically described on page 1 lines 19-20, And the specific minerals involved in these reactions have also been added on page 1, lines 19-20 as follows. Page 1 lines 19-20: "Groundwater chemistry is controlled by minerals (halite, gypsum, anhydrite, mirabilite) dissolution, silicate weathering, cation exchange, evaporation and minerals (halite, gypsum, anhydrite, aragonite, calcite, dolomite) precipitation"

Comment 3): The % of water from the systems (line 22) cannot be that precisely estimated.

Reply: The % of water from the systems are estimated using 2D groundwater flow model and have been rounded to a whole number as follows. Page 1 lines 24-25: "The quantity of water discharge from these three systems accounts for approximately 83%, 14% and 3%, respectively, of the total groundwater quantity of the watershed." In this study, a steady numerical model was constructed to present the groundwater flow pattern. Based on the simulated results of the groundwater flow characteristics, three different hierarchical groundwater flow systems can be divided in this study area, and the discharge area of each system can be identified. The discharge water quantity of each system can be gained from the corresponding cells of the simulated results of TOUGH2, and then quantitative percentages of water from the systems can be esti-
mated.

**Introduction Part:**

General comments: The introduction references a lot of literature, but needs more details. You should outline the specific issues that you are addressing in this paper more fully – it is good to mention a range of features (resources, ecology etc.) but the main purpose of this paper, which is to understand the hydrogeology, needs more emphasis. You should expand this section to explain in more detail how this work specifically addresses an important hydrogeology question and how it relates to our understanding of groundwater in these types of basins in general. HESS is an international journal and so papers need to appeal to readers working in other regions, so it is critical that you explain the general importance of the work. Perhaps refer to basins elsewhere and explain the common questions that this study will help to address.

Reply: The introduction has been revised to further illustrate main purpose of this paper as follows. Page 1 lines29-31 and Page 2 lines 1-2: "Closed basins in arid and semiarid areas (e.g. the Great Artesian Basin and Murray Basin in Australia, Mingin Basin and Qaidam Basin in China, Death Valley in United States) have been the focus of attention due to their water scarcity, fragile ecology and rich mineral resources related to salt lakes (Edmunds et al., 2006; Lowenstein and Risacher, 2009; Love et al., 2013; Shand et al., 2013; Stone and Edmunds, 2014; He et al., 2015; Cartwright et al., 2017; Love et al., 2017; Priestley et al., 2017a; Xiao et al., 2017)." Page 2 lines 17-24: "The systematic understanding of regional groundwater regimes is still inadequate. This would limit the comprehensive planning and management of groundwater and salt lake mineral resource exploitation, and finally make it difficult to safeguard the circulation of the groundwater system and maintain the eco-environment at balance. Therefore, several attempts have been made to understand the regional groundwater regimes (Tan et al., 2009; Gu et al., 2017; Xiao et al., 2017), but very little research reported the circulation and evolution of groundwater from the mountain pass area to the central terminal lake area due to the notable difficulties to move through and access the
swamps on the lacustrine plain. This would greatly limit the full understanding of the role of hydrogeological processes in the basin." Page 3 lines 2-7: "The combination of these approaches is robust to reveal groundwater origin, flow regimes, renewability, hydrochemical evolution, inter-aquifers mixing, as well as surface water and groundwater interactions, etc., in basins with complex hydrogeology or sparse monitoring data, and has been successfully applied in many basins such as the Great Artesian Basin and Murray Basin in Australia, Michigan Basin in US, Mingin Basin and Ordos Plateau in China, Stampriet Basin in Africa (Edmunds et al., 2006; Banks et al., 2010; Love et al., 2013; Stone and Edmunds, 2014; Su et al., 2016; Cartwright and Morgenstern, 2017; Love et al., 2017; Petts et al., 2017; Priestley et al., 2017b)." Page 3 lines 8-14: "The specific aims of the present study are to: (1) identify the recharge source of groundwater, (2) assess the regional groundwater chemistry characteristics, (3) determine the controlling mechanisms of hydrogeochemistry, (4) delineate regional groundwater flow patterns, (5) and ultimately establish systematic regional groundwater regimes from the mountain pass to the terminal lake in the typical Golmud watershed of Qaidam Basin. This study would provide insights into the origin, recharge environment, flow regime and geochemical evolution of regional groundwater in arid endorheic watersheds of Qaidam Basin, and provide reference for other arid closed basins in northwest China as well as similar endorheic watersheds worldwide."

Specific comments Comment 1): Page 2 lines 13-17: This just says that it was difficult to do the research; perhaps more importantly is some indication as to why this information would be useful.

Reply: The reasons for why the regional groundwater regime are added and emphasized in the page 2 lines 17-20. As following: Page 2 lines 17-20: "The systematic understanding of regional groundwater regime is still inadequate. This would limit the comprehensive plan and management of groundwater and salt lake mineral resources exploitation, and finally make it difficult to safeguard the fine circulation of groundwater system and maintain the eco-environment balance."

**HESSD**
Comment 2): Page 2 lines 18-27: Be specific with the term "isotopes" as there is a considerable difference between the information that you get from the stable isotopes (O & H) and radioactive isotopes such as 14C. Better to specific which ones.

Reply: The specified isotopes have been indicated on page 2 line 28-31. As following: "To achieve this aim, a comprehensive approach using environmental isotopes (2H, 18O, 3H, 13C, 14C) and hydrochemistry coupled with numerical simulation was performed. Stable hydrogen and oxygen can provide valuable information on the origin and recharge environment of groundwater, and radioactive isotopes such as 3H, 14C record the residence time of groundwater".

Comment 3): You need to develop the aims better. You can do this either by framing a hypothesis or by explaining the aims more fully. At the moment, you just say that there are some techniques that we can use to help us understand groundwater systems and you are going to apply them to this basin. What specifically do you hope to achieve and how will it inform the understanding of this basin and similar ones?

Reply: The main aim of this study was presented on page 2 lines 26-28. And the further explanation and what we hope to achieve are added on page 3 lines 8-14. As following: "The specific aims of the present study are to: (1) identify the recharge source of groundwater, (2) assess the regional groundwater chemistry characteristics, (3) determine the controlling mechanisms of hydrogeochemistry, (4) delineate regional groundwater flow patterns, (5) and ultimately establish systematic regional groundwater regimes from the mountain pass to the terminal lake in the typical Golmud watershed of Qaidam Basin. This study would provide insights into the origin, recharge environment, flow regime and geochemical evolution of regional groundwater in arid endorheic watersheds of Qaidam Basin, and provide reference for other arid closed basins in northwest China as well as similar endorheic watersheds worldwide."

Study area Part:

General comments: The study area section needs more detail. This is a hydrochem-
istry paper that as background requires an understanding of the hydrogeology. However, many details are lacking, such as: Comment 1): You should describe what is known about the flow system, for example where are the recharge and discharge areas?

Reply: The related info of the flow system has been added on page 4 lines 4-12. As follows: "Groundwater in the basin is mainly recharged by Golmud River seepage through riverbed in the alluvial fan and bedrock lateral inflow at the southern mountain front, and flows from alluvial fan in the south to the basin center in the north (Fig.1c). Much of groundwater overflows as springs at the front of the alluvial fan due to the aquifers lithology becoming finer in grain size. The depth to groundwater is less than 3 m in most areas from the front of alluvial fan to the basin center, resulting in evaporation loss of groundwater. The regional groundwater finally discharges to a terminal lake, and experiences loss though evaporation."

Comment 2): The maps should show recharge areas and groundwater flow paths.

Reply: The information concerning recharge areas and groundwater flow paths has been added to Fig.1c (in this Response file is Figure A) as follows.

Comment 3): What do we know about hydraulic properties (especially K)?

Reply: The lithology distribution and related hydraulic properties in the basin has been described on page 3 lines 29-31 and page 4 line 1 as follows. "The regional Quaternary aquifers in the basin vary from single unconfined gravel and sand layers with hydraulic conductivity (K) greater than 50 m/d in the alluvial fan to multi-layers of silt and clay with hydraulic conductivity (K) ranging from 0.1 m/d to 0.001 m/d in the low-lying depression (basin center)."

Comment 4): Without this information it is very difficult to understand the study and the statement "Overall, groundwater in the basin originates from Golmud River seepage and bedrock lateral flow in the alluvial fan, and topography results in flow towards the
low-lying depression (basin center)." Is hard to assess.

Reply: This part has been rewritten, the sedimentary lithology of aquifers from the alluvial fan in the south to the basin center in the north has been described on Page 3 lines 29-30. And the hydraulic properties (hydraulic conductivity K) has been simply introduced with the aquifers lithology introduce on page 3 lines 30-31. The influence of aquitards (clay layers) on groundwater flow has been told on page 4 lines 1-2. Recharge, discharge and flow paths of the groundwater system has been introduced on page 4 lines 6-12, and the schematic features have been added in Fig.1c. After the revision, readers may basically understand the hydrogeology of the study area.

Materials and Methods:

General comments: For the groundwater samples, you need more details on the wells. The interpretation of data from long-screened production wells is more difficult than from monitoring bores with short screens and no pumping. I do not see a data table in the paper (the tables are just summaries). It is critical that you provide the raw data (HESS will let you do this as a supplement). For the groundwater samples, you need to specify The well depths and typical screen intervals Whether these are monitoring or production wells The aquifer that is sampled The table also needs your geochemical data in it and details such as sampling date etc. The actual data is an important part of this study and must be made available with the paper.

Reply: We agree that scientific results and achievements should be shared and communicated. This is the reason we submitted this paper to HESS for public publication. Unfortunately there are several severe retrictions associated with the data. Although I have permission from a number of sources where the data was obtained to use the data for plots and modelling, I do not have permission to publish the data. To address this problem, these data has been presented in the statistical summary tables (Table 1 & 2) and in the Figures (Fig. 2, 3, 5, 6, 7). We think these presenting forms of data can efficiently solve the contradiction between original data confidentiality and the sharing
of scientific results, and also be useful and helpful for the study. In addition, if it is necessary, some other figures or statistical summary tables can be added in this paper. We sincerely hope this would not be the critical obstacle for publishing this paper in HESS.

Specific comment 1): The 14C analyses involved using a field precipitation technique. As discussed by Aggarwal et al. (2014: Groundwater, 52, 20-24) this is prone to errors by atmospheric contamination. Did you assess the possibilities that atmospheric contamination has occurred (using field blanks or repeated samples)? At the very least, you should discuss this.

Reply: We have carefully read the paper published by Aggarwal et al. (2014: Groundwater, 52, 20-24) and also consulted the Laboratory of Groundwater Sciences and Engineering in the Institute of Hydrogeology and Environmental Geology, Chinese Academy of Geological Sciences (IHEG-CAGS) (Shijiazhuang, Hebei Province, China) which did these 14C analysis. We are sure that our samples analysis results are valid. The reason are as following. It is agreed that 14C samples may be polluted by atmospheric CO2 during the sampling and analysis progress. Generally (the experiment of Aggarwal et al. (2014: Groundwater, 52, 20-24) was also in this case), the container for sampled groundwater is with an opening at the top for adding reagents for a relative long time. Only after all reagents were added, the opening was closed and a stirring rod, inserted through the cap, was used to ensure complete removal of dissolved carbon. Under this condition, the atmospheric CO2 may pollute the samples in the duration of adding reagent when the top is opening all the time and though the gap between the stirrer and the container cap when stirring (Figure B).

In our study, this issue has been considered. All the samples were sampled and prepared following the guidance of the Laboratory of Groundwater Sciences and Engineering in the Institute of Hydrogeology and Environmental Geology, Chinese Academy of Geological Sciences (IHEG-CAGS) (Shijiazhuang, Hebei Province, China). The laboratory improved the process of 14C sampling and analysis, which can efficiently reduce
or even eliminate the possibility of atmospheric CO2 pollution. The Sampling and precipitating instrument are schematically shown in Figure C. The sampling groundwater had pumped in to the container through the water inlet at the top. The container had been washed using the sampling groundwater for three times, and then the pumping tube was put at the bottom of the container for filling up the container. Till more than 10 L water overflowed from the top of container, closed the container water inlet shortly. The cap at the top would be opened in very short time (generally seconds) when adding reagents, and closed immediately after adding the corresponding reagents. In order to avoid atmospheric CO2 entering the container through the gap between stirrer and container cap, the stirrer was canceled, and the sealing container was put on the ground for shaking to replace the stirrer effect. All the field and laboratory pretreatment and analysis steps which may have change to occur atmospheric contamination had been improved and also has strict Quality Assessment. The Laboratory had done experiments to test the effectiveness of this improved method and proved it can effectively eliminate the effect of atmospheric CO2 pollution.

Specific comment 2): Page 4, line 22. It is not clear what you mean by "The standard deviation of analytical results between 0.7 pMC and 1.0 pMC" is this from repeat analyses or is it a typical range from the lab (and why the range of values?)

Reply: This previous expression is not correct, and it has been corrected according to the lab providing info as the precision of 14C activity being  $\pm 0.3\%$  (Page 5 lines 19-20).

Specific comment 3): As noted above, you need to include these data in a Table

Reply: Please refer the response of "Materials and Methods" section.

Specific comment 4): Section 3.2. There are many values in here but little indication as to where they come from. Some of these details need to be in Section 2 as they are part of the background understanding. Without a clear description of the hydrogeology and flowpaths it is difficult to assess the appropriateness of the modelling (recharge is mentioned here, but it has not been explained where the recharge areas are and what

**HESSD**
any prior estimates of recharge are).

Reply: A report providing these values was newly cited in this part. The Section 2 has been rewritten to provide detailed hydrogeology and flowpaths of the study area. Recharge info has been described in Section 2 and the recharge area has been shown in Fig.1 of the manuscript.

Section 4 Results:

Comment 1): Table 1 is only a summary table, we need the data!

Reply: Please refer the response of "Materials and Methods" section.

Comment 2): This section is a reasonable description of the data. However, it is a little brief in places and as discussed below, this does not help with the interpretation. I am not sure that defining water types is that useful as ion ratios are probably more use for understanding processes (it is a common but slightly outdated way of discussing hydrogeochemistry). While it is true that waters do "evolve" in composition from rainfall to brines (page 6, lines 13-20), that tells us little about the processes that cause them to do so.

Reply: The section has been rewritten. More detailed info about the surface water and groundwater chemistry has been added on page 7 lines 10-34 as follows. "Surface water and groundwater present distinct major solute chemistry across the study area. As shown in Table 1, the concentration of ions in RW demonstrates an increase along river flow paths, with a TDS values varying from 393 mg/L to 2,319 mg/L. The TDS value of FLW (L1) is much higher than that of RW in the low-lying depression (Zone 5), with the TDS value of 10,937 mg/L. While the salt lake waters (SLW) have extremely high TDS values ranging from 339,098 mg/L to 403,758 mg/L. The dominant ions of RW are HCO3- and Na+ with the concentration range of 184-215 mg/L for HCO3- and 63-92 mg/L for Na+, respectively, in the alluvial fan area (Zone 2), and gradually evolve to CI- and Na+ with the concentration range of 655-1,776 mg/L for CI- and 438-996

HESSD
mg/L for Na+, respectively, in the low-lying depression (Zone 5). FLW (L1) has the same dominant ions with RW in the low-lying depression (Zone 5), but with higher concentration of 5,912 mg/L for CI- and 2,957 mg/L for Na+. SLW is dominated by CI- and Mg2+ with the concentration range of 276,849 mg/L to 285,780 mg/L for CIand 99,500 mg/L to 100,240 mg/L for Mg2+, respectively. Overall, the surface water types evolve from HCO3ÂůCI-CaÂůMgÂůNa type in the alluvial fan area (Zone 2) to CI-Na, CI-K-Na and CI-Mg type in the low-lying central depression (Zone 5) (Figure 2a). Groundwater shows a similar hydrochemical evolution along the flow path. The average TDS values vary from 618 mg/L to 32,029 mg/L for SGW and from 547 mg/L to 1,401 mg/L for DGW from the upstream area (Zone 2) to the middle-lower stream area (Zone 4). DGW is much fresher when contrasted with the SGW at the same location (Figure 2c). There is essentially no difference in TDS between SGW and DGW from the central depression (Zone 5) with the values ranging from 336,229 mg/L to 361,200 mg/L for SGW and 370.940 mg/L for representative DGW (Table 1). Groundwater in the alluvial fan area (Zone 2) is dominated by HCO3-, CI- and Na+ with the concentration ranging from 89 mg/L to 309 mg/L for HCO3-, from 90 mg/L to 437 mg/L for CI-, and from 79 mg/L to 232 mg/L for Na+, respectively. To the middle-lower stream area (Zone 4), the dominant ions vary to CI- and Na+ for both SGW and DGW. The mean concentration of CI- is 11,550 mg/L for SGW and 263 mg/L for DGW, and the average concentration of Na+ is 10,464 mg/L for SGW and 407 mg/L for DGW. All groundwaters including SGW and DGW in the basin center (Zone 5) are dominated by Cl-, Na+ and Mg2+. SGW has the concentration ranging from 215,561 mg/L to 227,451 mg/L for CI-, from 12,388 mg/L to 35,713 mg/L for Na+, and from 53,480 mg/L to 64,860 mg/L for Mg2+. The concentration of representative DGW is 222,404 mg/L for Cl-, 32,378 mg/L for Na+, and 57,079 mg/L for Mg2+. Overall, the water types of both SGW and DGW evolve from HCO3ÂuCI-CaÂuMgÂuNa type in the upstream area (Zone 2) to CI-Na type in the middle-lower stream area (Zone 4), and eventually to CI-Mg type in the low-lying depression (Zone 5) (Figure 2b). "

Comment 3): There is some material here that is discussion and so belongs later (eg
page 6, lines 9-10) and some overly speculative conclusions (eg page 6, lines 19-20) that need to be properly justified or omitted. Conversely, there is some material in section 5 (such as the description of the Na/Cl rations) that should be here as this is where you present the data.

Reply: The overly speculative conclusions (eg page 6, line 24-25; page 6, line 9) has been deleted from this part, and some of them will be put into section 5 with proper justification. The deleted sentences are as following: Page 6, line 9 "This increase has been attributed to increasing evaporation (Wang et al., 2013)" Page 6, line 10-12 "implying intensive evaporation and potentially a complex hydrogeochemical history, as well as possible multi-water sources outside of Golmud River water (Lowenstein and Risacher, 2009)." Page 6, lines 23-24 "indicating evaporation has had a significant influence on the hydrochemical development of SGW." Page 6, lines 24-25 "implying extremely long residence times in aquifers and complete reaction with aquifer mediums."

Comment 4): This section needs restructuring so that all the data that you use to make interpretations is presented and described adequately.

Reply: This section has been reorganized and more detailed info about the surface water and groundwater chemistry has been added into to this part. We hope the new restructuring description is adequate to subsequent discussion. Please refer Section 4.1 and 4.2

Section 5 Discussion:

General comment: Perhaps due to the tendency to explain aspects of the study briefly, and a lack of primary data, there are several conclusions in this section that are questionable (or at least need more explanation).

Comment 1): The interpretation of the 14C residence times (Page 9) has several issues: Most importantly, you seem to have samples with low 14C but measurable Tri-
tium (Fig. 3). If so these must be mixtures between older water (low 14C, 3H free) and younger water (high 14C and high 3H) as the time required for measurable 14C to decay wipes out all the 3H. You cannot calculate residence times from such waters. A clear explanation of how the radioisotopes are behaving is required before you do any calculations.

Reply: It is truth that some samples are with low 14C activity but measurable tritium content (iijd1 TU). This is caused by the mixture with shallow phreatic water which has relative high tritium content in boreholes due to the poor sealing borehole structure. We have discussed this on page 8 line 31-33. These samples data have been discarded, and only tritium free samples are used for dating of the ancient groundwater.

Secondly, just saying that you applied the Tamers model or an unspecified statistical is inadequate. The correction of 14C ages is commonly difficult and needs more justification (there are numerous papers that address this in many basins). Even if in the end you just use a simple correction, you need to justify that you understand what is happening in the C-system and rule out possibilities such as open-system carbonate dissolution or methanogenesis. Also you need to outline what the % of dead carbon is and whether that is reasonable.

Reply: As only part of 14C data having carbon-13 data, we choose statistical and Tamers model to correct all the carbon-14 age. These two model can provide some reference or insight into the residence time of ancient groundwater in the watershed. However, they are of limited interest due to their simplification of geochemical reactions beyond the recharge area, and the assumptions of a fully closed system. Thus, a better approach using carbon-13 was applied to estimate tritium free groundwater age. The relative discussion has been added on page 9 line 4-23 as follows. "Radiocarbon activity of groundwater can be significantly influenced by geochemical reactions (e.g. carbon minerals dissolution, isotopic exchange processes) during subsurface infiltration and in the aquifers (Cartwright et al., 2010b). It is therefore essential to correct the 14C activity on the total dissolved inorganic carbon (TDIC) before using it for ground-

HESSD
water age estimation. Many model such as statistical models, geochemical models, and mixing models were proposed for 14C activity correction. Most of the models are of limited interest due to the assumptions of fully closed system or open system, simplification or even fully ignorance of geochemical reactions beyond the recharge area. Carbon-13 based model is a good approach to correct the influence of geochemical reactions on 14C activity on TDIC, and suitable for both open and closed system. The measured apparent 14C activity (14Cuncorr) on TDIC were corrected using  $\delta$ 13C as following (Clark and Fritz, 1997): ( 14)C corr = ( 14)C uncorr (  $\delta$ ( 13)C rech -  $\delta(^{13})C$  carb )/( $\delta(^{13})C$  TDIC -  $\delta(^{13})C$  carb ) Where 14Ccorr is the corrected 14C activity on TDIC,  $\delta$ 13CTDIC is measured  $\delta$ 13C ratio on TDIC,  $\delta$ 13Crech is the assumed initial  $\delta$ 13C ratio, and  $\delta$ 13C carb is the  $\delta$ 13C ratio of carbonate being dissolved. Groundwater in the study area is mainly recharged by Golmud River seepage in the upper alluvial fan located near parts of the Gobi desert where there is a lack of vegetation. The 14C activity and  $\delta$ 13C ratio on TDIC of the water would not be changed when infiltrating though the unsaturated zone. Thus, the  $\delta$ 13Crech ratio should be equal or close to the atmospheric value (-6.4  $\infty$ .  $\delta$ 13Ccarb is close to 0  $\infty$  (Clark and Fritz, 1997). Only some of the tritium free DGW samples in Zone 3 and Zone 4 have measured  $\delta$ 13C data, and these were selected to calculate groundwater age using the aforementioned  $\delta$ 13C correction approach. The age of DGW in Zone 3 and Zone 4 ranges from 2,264 years to 20,754 years along the flow paths. Due to the absence of radiocarbon data, the age of paleo groundwater in Zone 5 cannot be calculated, but it is certain that the age is more than 20,000 years which was deduced from the oldest age of groundwater in Zone 4 (20,754 years)."

The Tamers model implicitly assumes that only carbonate dissolution occurs. However, in moderately saline groundwater, you may have carbonate precipitation. Have you assessed this?

Reply: This has been considered and highly saline samples were not used. As shown in Fig. C, the northern most 14C sample is taken from the middle area of Zone 4. It

**HESSD**
has a TDS value of 1090 mg/L. All the other samples used for 14C age calculating are fresh water (

a high tritium content of 18.9 TU, presenting modern water isotopic signatures. This may be caused by the mixture with the infiltrating modern surface water. DGW in the overflow zone (Zone 3) and middle lower stream area (Zone 4) are with tritium content ranging from

is close to 0 ‰ (Clark and Fritz, 1997). Only some of the tritium free DGW samples in Zone 3 and Zone 4 have measured  $\delta$ 13C data, and these were selected to calculate groundwater age using the aforementioned  $\delta$ 13C correction approach. The age of DGW in Zone 3 and Zone 4 ranges from 2,264 years to 20,754 years along the flow paths. Due to the absence of radiocarbon data, the age of paleo groundwater in Zone 5 cannot be calculated, but it is certain that the age is more than 20,000 years which was deduced from the oldest age of groundwater in Zone 4 (20,754 years)."

Comment 2): The interpretation of the evaporation based on the stable isotopes (page 8, lines 15-17) seems to imply that evaporation is occurring along the flow paths. However, evaporation is something that can only occur at or near the surface and not directly from deeper groundwater. Do you mean that different degrees of evaporation occurs in different areas during recharge? If so, how does that fit with your conceptualization of the hydrogeology?

Reply: It is sure that evaporation can only occur at or near the surface. In this part, we do not mean all groundwaters regardless the depth were influenced by evaporation. We only say that shallow phreatic groundwater from the overflow zone to the downstream area has been influenced by evaporation. We have added some new interpretation in the manuscript to make it clear. The detailed responses are as following. Page 10, line 30: "The  $\delta$ D and  $\delta$ 18O values of the SGW and DGW demonstrate different varying trends along the groundwater flow path." This sentence only tells readers that SGW and DGW have different varying trends of stable hydrogen and oxygen along the flow paths, we do not mean that both SGW (shallow phreatic groundwater) and DGW (deep confined groundwater) samples plot along the evaporation line and are influenced by evaporation. The specific varying trends of stable isotopes and their interpretation has been given on page 10 lines 30-31, page 11 lines 1-4 and 7-11 for shallow phreatic groundwater, Page 11 lines 14-26 for deep confined groundwater. Page 10, lines 30-31 and page 11 line 1: "The SGW shows a gradual positive enrichment trend in heavy isotopes along the LEL (Fig.5c), implying the influence of evaporation." In this sentence,

**HESSD**
we point out that the shallow phreatic groundwaters plot along the local evaporation line (LEL) and present a gradual enriched trend along the flow paths. Thus we can conclude that evaporation has influenced the shallow phreatic groundwater along the flow paths. We discussed the stable water isotopic signatures of groundwater in the alluvial fan area on page 11 lines 1-4, and point out that groundwater here is directly recharged by river water and has a short residence time in the aquifers. The similar stable isotopes features of groundwater here with that of recharged water also indicates that evaporation has nearly no influence on groundwater in the alluvial fan, this is consistent with the greater depth of groundwater in the piedmont area. In order to express this more clearly, we changed the sentence as following: "For the alluvial fan (Zone 2), the  $\delta D$  and  $\delta 18O$  values of groundwater are very similar to that of river water in the alluvial fan (Zone 2) and groundwater in the mountainous area (Zone 1) (Table 2), indicating groundwater in the alluvial fan (Zone 2) is recharged directly by the seepage of river water and lateral inflow from the mountainous area, and out of the influence of evaporation." (In revised manuscript, Page 11 lines 1-4.) Shallow phreatic groundwater in the overflow zone (Zone 3) and the middle-lower stream area was discussed on page 11 lines 7-11. We also added some interpretations about the evaporation influence on page 11 lines 9-11. The related sentences are as following: "SGW in the overflow zone (Zone 3) and the middle-lower stream area (Zone 4) has relative stable water isotope values compared with that in the alluvial fan and plots along the LEL, indicating SGW is influenced by evaporation from the overflow area (Zone 3) to the downstream. SGW in these two zones (Zone 3 & 4) also presents similar stable hydrogen and oxygen isotopic signatures as the river waters in the same area (Table 2), implying SGW has a very close hydraulic relationship with the rivers." (In revised manuscript, Page 11 lines 7-11.) The stable water isotopic signatures of deep confined groundwater (DGW) are described and interpreted on page 11 lines 14-15. DGW shows a gradual depleted trend along the flow paths from the alluvial fan (Zone 2) to the middle-lower stream area (Zone 4). It does not present a gradual enriched evolving trend along the flow paths. We do not express that DGW in the study area is influenced by evaporation.

**HESSD**
Comment 3): In several places, the similarity or differences in the geochemistry are mentioned, but there are no attempts to quantify this (the reader has to basically look at the figures or tables and make their own assessment). At the very least put the ranges and differences in means in the text, but preferably try to use something like PCA or ANOVA to better justify this.

Reply: We have put the ranges and values, which were involved into the discussion, in the paper (Page 12 lines 24-26, and Page 13 lines 4-5, 11-12 etc.). It should be clear with these quantitative values in text, and intuitive with the figures.

Comment 4): Section 5.2 on the hydrogeochemistry also has several issues. The first part of the section is really just an (old) textbook introduction. Yes, these processes control the geochemistry, but the details are more subtle than this. Figure 6 is just a broad generalization and while it is a useful conceptualization, it does not tell us much about specific processes (objective here)

Reply: In the first paragraph of this secction, the Gibbs diagrams have been introduced to identify the overall mechanisms controlling hydrogeochemistry. We use the first sentence to briefly introduce the primary controlling processes of natural groundwater chemistry, and the other sentences are specific analysis using the Gibbs diagrams in different zones of the study area. Although the Gibbs diagrams method cannot tell the specific processes occurred in aquifers, they can give us an intuitive understanding of the main natural mechanisms controlling the groundwater chemical composition. If only specific processes analysis was used in the text, readers may feel the processes too complicated and have no intuitive and overall impression. Thus, this method has been widely used in groundwater chemistry researches. After this diagram, the relationship between major ions are compared to constrain the specific processes controlling groundwater chemistry. In general, we suggest to keep the Gibbs diagrams analysis.

The subsequent statements on lines 22-30 are unjustified. You need to relate this discussion back to the description of the geochemistry and explain specifically how

HESSD
you came to these conclusions (ie what in the hydeogeochemistry tells us that waterrock interaction has occurred, what minerals are dissolving / precipitating etc). At the moment your interpretation just relies on where in the system the water is from (this might be correct but it does not make use of any of the geochemistry).

Reply: The truth is that we get the conclusions from the Gibbs diagrams, not relying on where in the system the water is from. From the Gibbs diagrams, we can identify which groundwaters are controlled by water-rock interaction and which by evaporation-mineral precipitation. We analyze why these processes primarily controlling the groundwater chemical composition based on the hydrogeological condition on page 12 lines 8-12, and 13-15. Page 12 lines 8-9: "Water-rock interaction processes dominant the controls on groundwater chemistry at all depths in Zone 2 due to the great depth and the negligible impact of evaporation." Page 12 lines 9-12: "For the overflow zone (Zone 3) and the middle-lower stream area (Zone 4), the governing mechanisms for SGW change from water-rock interaction to evaporation-mineral precipitation due to the gradual decrease of groundwater depth and recharge inputs from waters having undergone the influence of intensive evaporation in that part of the basin." Some special DGW samples plotted in evaporation-crystallization domain are briefly interpreted based on the TDS values as following. Page 12 lines 13-15:"Two DGW samples are observed to plot in the evaporation-crystallization domain (Fig.6). This is due to a high TDS and over-saturation of evaporative minerals (such as Aragonite, Calcite and Dolomite) in the groundwater resulting in mineral precipitation (crystallization)." The specific explanation based on the geochemistry are in the following paragraphs of the manuscript.

Na/CI ratios are nor definitive in constraining halite dissolution. Rainfall has Na/CI ratios that are close to 1 (generally 0.7-1.2) and given that ion exchange may also occur, you cannot distinguish evaporation and halite dissolution. Really you need Br and to look at CI/Br ratios as halite dissolution produces CI/Br ratios that are orders of magnitude higher than halite dissolution. The SI indices are not relevant.
Reply: We agree that Cl/Br ratio is a good way to constrain the halite dissolution. Unfortunately, we do not have the Br data in the current study. In our future study, we will consider to obtain Br data to further justify the halite dissolution. It truth that we cannot determined the halite dissolution processes only by Na/Cl ratios. In our text, we say "halite dissolution is potentially a primary process/source of Na+ and CImineralization in groundwater. (Page 12, lines 20-21)" from the Na/Cl ratios. The SI of halite also shows that halite is under saturation, suggesting halite can be dissolved and enter into groundwater. In addition, the core drilling also demonstrates halite is widespread in the aquifer materials, and we have provided this info on page 3 lines 28-29 and Page 12 line 22-24. We believe that all the three evidences above can prove that halite is one of the sources of Na+ and CI- in groundwater. Rainfall can provide a Na/CI ratio close to 1, however groundwater shows a gradual increasing concentration of Na+ and Cl-, this must be caused by some other processes. Evaporation could be an important reason to proportionally increase the Na+ and Cl-, but this influences can only occur at or near the ground surface. To deep groundwater, their proportional increase of Na+ and CI- should be the result of halite dissolution. Comparing the increase rate of these two ions in different depth along flow pates, we can clearly see that shallow groundwater has a higher increase rate, and this is due to the evaporation concentration effect. Although evaporation has a greater influence on the increase of Na+ and CI- concentration in shallow groundwater, it is sure that halite dissolution also contributes to the increase of Na+ and CI- along the flow pate. Ion exchanges can also influence the Na/CI ratio, but it would not proportionally change the Na+ and CI- concentration. The related discussions of the ion exchanges are on page 12 lines 30-32 and Page 13 lines 4-6.

The explanation of the geochemical processes on page 10 would be helped if the mineralogy of the aquifers had been properly described (which of any these minerals exist-that is obviously important).

Reply: The mineralogy of the aquifers had been described on page 3 lines 28-29

**HESSD**
and Page 12 line 22-24 as following. Page 3 lines 28-29: "Core drilling records also show many salt-bearing deposits such as halite, calcium, sulfate, sodium sulfate were observed throughout the strata" Page 12 line 22-24 as following: "In addition, core drilling demonstrated that evaporate salts such as halite, calcium sulfate and sodium sulfate are widespread in the aquifer materials, and can provide the solute source."

Finally, it is not clear why the authors have looked at the geochemistry in this much detail. While understanding the geochemistry is important, it should inform a broader understanding of the system, for example: does it constrain inter-aquifer mixing or where the water was recharged, is it useful for interpretation of 14C ages, is there a palaeowater signal in the major ions as well in the stable isotopes that could be useful in detecting climate influences elsewhere. As it is, this section stands alone and is actually not well integrated to the study.

Reply: In the present study, a conceptual model of groundwater flow and hydrochemial evolution was established using the environmental isotopes, hydrogeochemistry, and numerical modelling. The stable isotopes was mainly used to reveal the water sources and recharge environment, as well as the relation between river and groundwater. Hydrogeochemistry is also be used to justify some of these interpretation such as water sources and river/groundwater relation (e.g. Page 11 line 4). The main aim of hydrogeochemistry is to reveal the groundwater chemical characteristics and its controlling mechanisms in groundwater flow system. This is an important part for understanding the regional conceptual model. The numerical modelling was used to reveal the flow regimes of groundwater. The Section 5.3 is not alone. In this section, a conceptual model of groundwater origin, flow and chemical evolution was established as shown in Fig. 8 based on the environment isotopes, hydrogeochemistry and numerical modelling research results. This could improve the understanding of groundwater flow and evolution regimes, and provide fundamental information for coping with the future issues such as water conflicts, salt lake exploitation and climate warming in the basin. This research can also provide references for understanding the hydrogeological pro-

**HESSD**
cesses in other similar endorheic watersheds of northwest China and elsewhere in the world.

**Conclusions:**

Comment: This is a reasonable summary of the main findings of the paper. However, as with the introduction, it needs a couple of extra paragraphs to explain the relevance to researchers working elsewhere (what perhaps have you done differently / better to other studies, are there any general points in understanding basins that you can make?). This will give the paper considerably more impact.

Reply: The relevant parts have been added on page 15 lines 9-14. And the last paragraph (page 16 lines 15-19) can also indicate the relevance of this research to other researchers in and other similar closed basin in the world. The relevant lines are as following. Page 15 lines 9-14: "Previous studies on arid closed basins such as the Great Artesian Basin, Murray Basin, Death Valley and Mingin Basin have established a lot of typical groundwater circulation and evolution regimes. While the Qaidam basin, a typical arid sedimentary closed basin formed with the uplift of the Tibetan plateau, has groundwater circulation patterns characterized by the complex tectonic activities, paleo climate variation, arid climate characteristics, sedimentary lithology, and systematic evolution from fresh to salt water. Studies of this basin can enhance the understanding of groundwater origin, flow regime and hydrogeochemical evolution in such complex tectonic influenced arid sedimentary closed basins worldwide." Page 16 lines 15-19: "This study enhanced the understanding of the origin, flow pattern, hydrochemical evolution and controlling mechanisms of the regional groundwater systems in the Qaidam Basin. These results can provide fundamental information for coping with future issues such as water conflicts, salt lake exploitation and climate warming in the basin, and also provide references for understanding the hydrogeological processes in other similar endorheic watersheds of northwest China and elsewhere in the world."

Please also note the supplement to this comment:
https://www.hydrol-earth-syst-sci-discuss.net/hess-2017-647/hess-2017-647-AC2-supplement.zip

**HESSD**

---

## Author Response (AR1)

**Cover letter to Editor**

Dear Editor,

Thanks for your help in the review process. We have finished the revision of our manuscript according to your comments. In this stage, two places were asked to add references and they have been finished as follows.

*(Page 3, lines 30-32 and Page 4, lines 1-2)* The regional Quaternary aquifers in the basin vary from single unconfined gravel and sand layers with hydraulic conductivity (K) greater than 50 m/d in the alluvial fan to multi-layers of silt and clay with hydraulic conductivity (K) ranging from 0.1 m/d to 0.001 m/d in the low-lying depression (basin center). Three continuous aquitards (clay layers) are found in the basin at depths of 60 m, 290 m and 450 m, respectively (Figure 8), which have significant influences on confining groundwater flow (Shao et al., 2017).

*(Page 4, lines 5-12)* The potential evaporation is extremely high (>2600 mm per year). This hyper-arid climate results in aquifers in the basin that do not obtain effective recharge from the local precipitation. Groundwater in the basin is mainly recharged by Golmud River seepage through the riverbed in the alluvial fan and bedrock lateral inflow at the southern mountain front, and flows from the alluvial fan in the south to the basin center in the north (Figure 1c). Much of groundwater overflows as springs at the front of the alluvial fan due to the fining of sediments in the aquifers downdip. The depth to groundwater is less than 3 m in most areas from the front of alluvial fan to the basin center, resulting in significant potentially evaporate loss of groundwater. The regional groundwater finally discharges to the terminal lake, and undergoes large evaporate loss (Shao et al., 2017).

*[Reference: Shao, J., Cui, Y., Xiao, Y., Li, Y., and Zhao, D.: Groundwater cycle pattern and groundwater resource evaluation in Golmud watershed of Qaidam Basin, China University of Geosciences (Beijing), 2017.]*

If you have any further comments or queries, please contact us.